# Enrichment of human IgA-coated bacterial vesicles in ulcerative colitis as a driver of inflammation

Himadri B. Thapa [1], Christina A. Passegger[2], Dominik Fleischhacker[1], Paul Kohl[1], Yi-Chi Chen [3], Ratchara Kalawong[3], Carmen Tam-Amersdorfer[2], Michael R. Gerstorfer[4], Jana Strahlhofer[1], Kristina Schild-Prüfert [1], Ellen L. Zechner [1,5,6], Andreas Blesl[7], Lukas Binder[7], Georg A. Busslinger [4], Leo Eberl [3], Gregor Gorkiewicz [5,8], Herbert Strobl [2,5], Christoph Högenauer [5,7] ✉ & Stefan Schild [1,5,6,9] ✉

The gut microbiome contributes to chronic inflammatory responses in ulcerative colitis (UC), but molecular mechanisms and disease-relevant effectors remain unclear. Here we analyze the pro-inflammatory properties of colonic fluid obtained during colonoscopy from UC and control patients. In patients with UC, we find that the pelletable effector fraction is composed mostly of bacterial extracellular vesicles (BEVs) that exhibit high IgA-levels and incite strong pro-inflammatory responses in IgA receptor-positive (CD89+) immune cells. Biopsy analyses reveal higher infiltration of CD89+ immune cells in the colonic mucosa from patients with UC than control individuals. Further studies show that IgA-coated BEVs, but not host-derived vesicles nor soluble IgA, are potent activators of pro-inflammatory responses in CD89+ cells. IgA-coated BEVs also exacerbate intestinal inflammation in a dextran sodium sulfate colitis model using transgenic mice expressing human CD89. Our data thus implicate a link between IgA-coated BEVs and intestinal inflammation via CD89+ immune cells, and also hint a potential new therapeutic target for UC.

Ulcerative colitis (UC), a chronic inflammatory bowel disease (IBD), is an immune-mediated disease affecting the colonic mucosa, showing continuous mucosal inflammation of the large intestine[1,2]. The course of clinical disease is characterized by periodic flares alternating with episodes of lower or absent disease activity or by chronic active disease without remission phases[3]. The worldwide incidence of UC rises continuously with currently 5 million estimated patients, yet its etiology remains poorly defined[4]. Despite recent advances in the treatment of UC with immunomodulatory drugs, considerable numbers of patients fail to respond to therapy and ultimately require total proctocolectomy[2,3,5,6]. Therefore, there is clinical demand for new therapeutic approaches in the treatment of UC[3,4,6], which requires a better understanding of the disease.

UC is connected with a loss of intestinal barrier integrity resulting in translocation of luminal content and activation of aberrant immune responses including proinflammatory T-cells, especially in genetically susceptible patients[6,7]. The intestinal microbiota is considered to play a pivotal role in the emergence of inflammation[8-11]. Alterations in the

[1]Institute of Molecular Biosciences, University of Graz, Graz, Austria. [2]Division of Immunology, Otto Loewi Research Center for Vascular Biology, Immunology and Inflammation, Medical University of Graz, Graz, Austria. [3]Department of Plant and Microbial Biology, University of Zurich, Zurich, Switzerland. [4]Research Center for Molecular Medicine (CeMM) of the Austrian Academy of Sciences, Vienna, Austria. [5]BioTechMed, Graz, Austria. [6]Field of Excellence Biohealth – University of Graz, Graz, Austria. [7]Division of Gastroenterology and Hepatology, Department of Internal Medicine, Medical University of Graz, Graz, Austria. [8]Diagnostic and Research Institute of Pathology, Medical University of Graz, Graz, Austria. [9]Austrian Agency for Health and Food Safety (AGES), Institute for Medical Microbiology and Hygiene, Graz, Austria. ✉e-mail: christoph.hoegenauer@medunigraz.at; stefan.schild@uni-graz.at

taxonomic and metabolic composition of the intestinal microbiome are well-established features in patients with UC[6,7,12]. However, knowledge of specific bacterial effectors leading to inflammation in UC is still very limited. With each individual bearing a unique microbiota, identifying bacterial taxa driving UC is a challenge[13]. It is becoming increasingly evident that UC does not follow the Koch's postulates as neither a single bacterium nor a specific microbial group are sufficient to cause the disease, but rather microbial dysbiosis combined with an aberrant immune response give rise to UC[4,8].

The complex crosstalk occurring between host and gut microbiota remains poorly understood. Microbes can modulate host physiology and immune responses by releasing myriad effectors that range from single molecules to large multi-factorial complexes. Generally, effectors can be recognized by various host sensors, known collectively as pattern recognition receptors (PRR). PRR comprise a variety membrane-bound and cytoplasmic types, which ultimately activate diverse networks of cellular responses. In this context, bacterial extracellular vesicles (BEVs) are a potentially important class of effectors. BEVs consist of microbial surface components as well as cytoplasmic or periplasmic components, which become trapped in their lumen during vesiculation[14,15]. Thus, BEVs are comprised of immunogenic compounds, including lipopolysaccharide (LPS), lipoteichoic acids (LTA), peptidoglycan, proteins, signaling molecules, and nucleic acids[16]. Release of these spherical, non-living nanoparticles is highly conserved among all bacterial species, and several BEV biogenesis pathways are triggered by stressors present in vivo. Consequently, BEV production is stimulated in associations with the host[14,15].

In this study we analyze colonic fluid samples from UC patients and non-IBD controls to identify factors with pro-inflammatory properties that are associated with UC. Our results show that the colonic BEV fractions of UC patients are characterized by high IgA-titers, high IgA-receptor (CD89)-IgA complex levels, and strong CD89-dependent inflammatory responses, which correlate with the extent of colonic inflammation in patients. Together our findings suggest a link between colonic bacterial-derived EVs, host-derived IgA and CD89⁺ immune cells, which are likely to cooperatively drive inflammatory responses in UC patients. Besides living bacteria of the intestinal microbiota also their secreted BEVs, which are non-living facsimiles of the donor bacteria, should be considered as inflammatory modulators in UC. Our findings not only offer a mechanistic explanation for chronic inflammation in UC patients, but suggest that abandoned B cell-dependent therapeutic approaches should be reevaluated in UC.

## Results

### Fractionation of colonic luminal samples derived from UC patients and non-IBD controls in soluble and pelletable effectors

We asked whether pro-inflammatory effectors are present in colonic fluid in association with UC. Stool harbors substantial amounts of indigestible material and has variable quality due to individual diet and unpredictable turn-around times until processing. In contrast colonic luminal fluid is cleaner due the patient's bowel preparation and fasting prior to colonoscopy and in practice, can be directly processed. Furthermore, mucosal washing during colonoscopy also collects microbial samples that are attached to the colonic mucosal surface[12]. Colonic luminal samples from UC patients and non-IBD controls were collected during colonoscopy (Fig. 1a, b). Patients lacking clinical signs of intestinal inflammation (28 of 74) were assigned to the non-IBD control group. Of the 46 patients with preexisting UC diagnosis, 30 were classified to be in endoscopic remission (inactive UC group, Mayo endoscopic subscore 0 or 1) and 16 to be in an active phase of colitis (active UC group, Mayo endoscopic subscore 2 or 3). Detailed inclusion and exclusion criteria of the particular study groups are provided in the method section. The three groups were comparable regarding age, sex, weight, and height, only the Ottawa Bowel Preparation Quality Scale (OBPQS)[17] indicated a lower quality of colonic

preparation in active UC patients consistent with a previous report[18]. Although the samples were processed within 1 h and kept on ice to minimize metabolic activity, we cannot completely rule out degradation or modifications in the colonic fluid during this time period.

Colonic luminal samples were then centrifuged and sterile-filtered to remove cells and debris. Each filtrate containing putative pro-inflammatory factors was then fractionated by ultracentrifugation to obtain supernatant – containing the soluble effector fraction (SEF) – and a pellet comprising the pelletable effector fraction (PEF). We observed high inter-individual variation in the protein biomass present in SEF and PEF for all three groups (Fig. 1c, d). The SEFs obtained from active UC group contained significant higher amounts of protein than those from the other groups, which might be due to the elevated OBPQS indicating more fecal residues or epithelial barrier impairment. In contrast, PEFs of all groups contained similar median protein biomasses. We did not observe variation in PEF yield according to flush volume during colonoscopy or OBPQS (Supplementary Fig. 1a, b). As an expected outcome of the ultracentrifugation step, the PEFs of all groups contained substantial quantities of spherical nanoparticles likely representing extracellular vesicles (EVs) derived from bacteria or the host with median values around $7 \times 10^{10}$ particles ml⁻¹ per µg protein equivalent (Fig. 1e). The number and diameter of nanoparticles were similar between groups. Mean and mode diameters around 160 nm and 90 nm are consistent with reported sizes of bacterial-derived extracellular vesicles (BEVs) as well as with host-derived extracellular vesicles (hEV) (Fig. 1f). Moreover, transmission electron microscopy revealed presence of EVs with typical morphology in two representative PEF samples of non-IBD controls and active UC patients (Supplementary Fig. 2). The PEFs of all three groups were further analyzed for EV parameters such as lipid and nucleic acid content[19] and similar amounts were found in all three groups (Supplementary Fig. 3a, b).

We determined the origin of the spherical nanoparticles by dot blot analyses allowing detection of CD9, CD81, and CD63 as markers for hEVs, ELISA for quantification of LTA for Gram-positive BEVs, and LAL assays to assess endotoxin [LPS] content for Gram-negative BEVs (Supplementary Fig. 3)[20,21]. The results confirmed the co-presence of all three EV types in PEFs of every group. BEVs and hEVs isolated from laboratory cultures served as internal standards and provided a rough estimate of the respective EV quantities in the human samples. Specifically, hEVs were purified from the intestinal epithelial cell line HT-29, Gram-positive BEVs from *Lactobacillus acidophilus*, and Gram-negative BEVs from *Bacteroides fragilis* and enterotoxigenic *Escherichia coli* (ETEC).

For each dot blot assay detecting CD9, CD81 or CD63, the signal intensity from a known amount of HT-29 derived hEVs was compared to the signal intensities of the human patient samples to estimate the percentage of hEVs in each PEF (Supplementary Fig. 3e–h, see methods for details). The median percent comprised by hEV in fractions of the non-IBD ranged from 0.8 to 10.2%, of the inactive UC from 0.1 to 28.1% and of the active UC group from 0.8 to 25.6% with highest levels based on CD81 and lowest levels based on CD63. The slight, but insignificant trend to more hEVs in UC vs. non-IBD samples is in line with previous studies that reported elevated hEV release upon stress stimuli including inflammation, which may also appear in UC[22–25]. The median levels for BEV components per µg PEF were not significantly different between the three human group groups ranging from 55.9 to 96.1 ng ml⁻¹ for LTA and 1135.2 to 1505.3 EU ml⁻¹ for LPS reactogenicity (Supplementary Fig. 3c, d). On average 1 µg of *L. acidophilus* BEVs corresponds to 1078.3 ng ml⁻¹ LTA, while 1 µg of *B. fragilis* and ETEC BEVs ranged from 1015.9 to 8409.4 EU ml⁻¹ LPS (Table 1). The wide range for the latter likely reflects the individual reactogenicity of different LPS species with *E. coli* LPS being the gold standard showing high biological activity[26,27]. Using the LTA content of *L. acidophilus* BEVs and a mean LPS reactogenicity of 4700 EU ml⁻¹ per µg EV the PEF

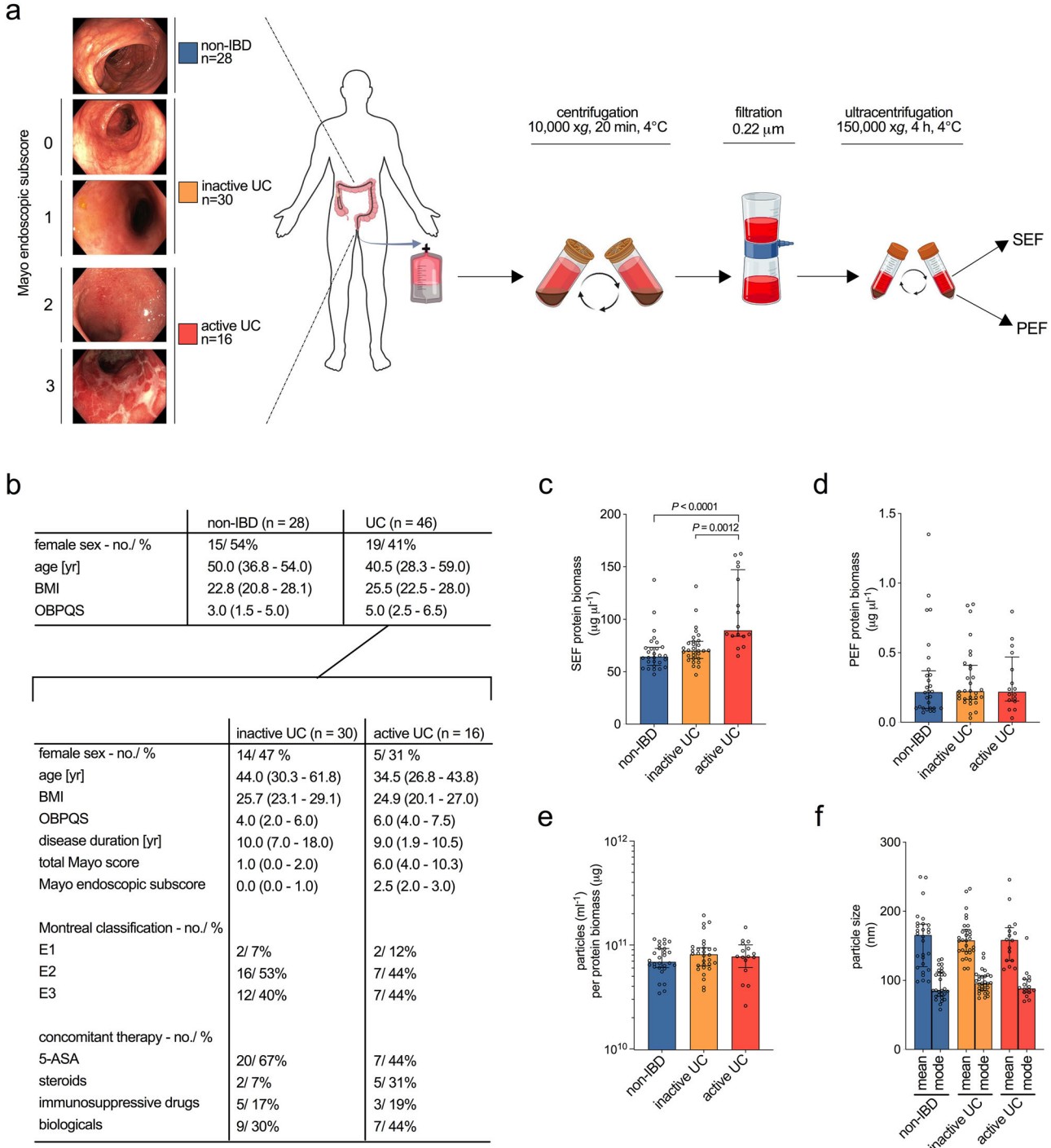

**Fig. 1 | Summary of human group assessment, colonic luminal sample processing and characterization of effector fractions. a** Classification of the enrolled patients ($n = 74$) in the study groups: non-IBD ($n = 28$), inactive UC ($n = 30$) and active UC ($n = 16$) is illustrated as well as colonic luminal fluid processing to the final soluble and pelletable effector fractions (SEF and PEF). Created in BioRender. Fleischhacker, D. (2025) https://BioRender.com/y93dulo. **b** Relevant clinical parameters of the non-IBD control group and UC group, which was further divided into the inactive and active UC groups based on Mayo endoscopic subscore. Data is presented as absolute number/ percentage of patients or as median (25th–75th percentile). Abbreviations as follows: BMI body mass index, OBPQS Ottawa Bowel Preparation Quality Scale, 5-ASA 5-aminosalicylic acid. **c**, **d** Total protein biomass determined by Bradford for the isolated

soluble and pelletable effector fractions (SEF and PEF) of the three groups, i.e., non-IBD (blue), inactive UC (yellow), and active UC (red). PEF values are presented as the protein biomass back-calculated to the original colonic luminal samples to eliminate the 100-fold concentration during fractionation. **e** Nanoparticle amount determined by nanoparticle tracking analysis (NTA) for the pelletable effector fractions of the three groups, i.e., non-IBD (blue), inactive UC (yellow) and active UC (red). **f** Mean and mode nanoparticle diameters determined by nanoparticle tracking analysis (NTA) for the pelletable effector fractions of the three groups, i.e., non-IBD (blue), inactive UC (yellow) and active UC (red). **c**–**f** Data is presented as median ± interquartile range. Statistical difference between the three groups was evaluated by Kruskal–Wallis with Dunn's test. **b**–**f** Source data are provided as a Source Data file.

**Table 1 Qualitative and quantitative analyses of the isolated EVs used in this study**

| Extracellular vesicles (EVs) derived from | Bradford (µg µl⁻¹) | Size – Mean (nm) | Size – Mode (nm) | Particle amount | SYTO-9 (ng NA µg⁻¹) | FM4-64 (RFU µg⁻¹) | LAL (EU µg⁻¹) | LTA (ng µg⁻¹) | IgA coating [IgA µg ml⁻¹ per protein biomass (µg)] |
|---|---|---|---|---|---|---|---|---|---|
| *Bacteroides fragilis* | 7.0 ± 0.3 | 166.9 ± 1.7 | 132.5 ± 7.0 | $2.39 \times 10^{11} \pm 1.01 \times 10^{11}$ | 19.3 ± 4.1 | 185929 ± 5412.7 | 3800.7 ± 1015.9 | NA | 89 ± 11 |
| *Lactobacillus acidophilus* | 2.4 ± 0.0 | 193.1 ± 8.0 | 126.2 ± 9.9 | $6.64 \times 10^{10} \pm 2.47 \times 10^{10}$ | 7.0 ± 0.14 | 7023 ± 355.2 | NA | 1078.3 ± 257.0 | 99 ± 6 |
| enterotoxigenic *Escherichia coli* (ETEC) | 8.3 ± 1.6 | 147.7 ± 4.5 | 121.0 ± 10.8 | $2.47 \times 10^{11} \pm 7.21 \times 10^{10}$ | 15.2 ± 1.6 | 118859 ± 2575.3 | 8409.4 ± 736.6 | NA | 97 ± 28 |
| intestinal epithelial cell line HT-29 | 14.2 ± 5.4 | 131.1 ± 4.0 | 108.2 ± 10.4 | $4.05 \times 10^{9} \pm 8.78 \times 10^{9}$ | 21.6 ± 3.9 | 12963 ± 1766.2 | NA | NA | <0.01 |

BEV preparations derived from lab-cultivated bacterial strains or hEV preparations derived from intestinal epithelial cells (HT-29) were analyzed for total protein biomass (determined by Bradford), mean and mode particle size and nanoparticle amount (determined by tracking NTA), lipid content (determined by FM 4-64 assay) and nucleic acid (NA) content (determined by SYTO 9 staining). In addition, LPS reactogenicity was determined by LAL assay for Gram-negative BEVs and LTA biomass was quantified for Gram-positive BEVs. Finally, efficiency of human IgA-coating is provided. Data is presented as mean ± standard deviation (n = 3).

of the patient groups roughly contained 30–40% BEVs. It should be noted, however, that this is only a rough approximation as (i) the LTA antibody might not detect all Gram-positive BEV species, (ii) LPS derived from diverse species shows different reactogenicity in LAL assays, and (iii) bacterial explosive membrane vesicles enriched with cytoplasmic content may not be adequately detected by these assays[15,26]. Thus, the predicted amounts of BEVs are likely an underestimation. Nonetheless, the quantification highlights that the PEF mainly consist of BEVs followed by host-derived hEVs, summing up to 50–80% of the total protein biomass. In addition to undetectable microbial vesicles, the remaining fraction could contain undigested food, intestinal mucus, fibers, flagella, and pili. However, TEM analyses of human-derived EV fractions showed no obvious bacterial flagellar- and pili-like structures (Supplementary Fig. 2). The distribution in the PEF samples is similar to a recently reported state-of-the-art protocol for EV isolation from human stool[28]. Thus, we will refer hereafter to the PEF as the EV fraction to reflect its main component.

**Analyses of colonic luminal samples identify elevated IgA-levels and pro-inflammatory signatures in EV fractions of UC patients**
Human intestinal HT-29 cells were exposed to individual SEF and EV fractions of all groups and analyzed for inflammatory cytokine IL-8 release (Fig. 2a). Notably, in all cell culture experiments we used SEF and EV concentrations below their levels in original colonic fluid (Fig. 1c, d) thereby likely corresponding to quantities perceived by host cells (see methods for detail). Exposure to EV fractions generally stimulated a higher median release of IL-8 compared to an equivalent protein biomass of SEFs. There was, however, no significant difference in cytokine release induced by either SEF or EV fractions of the inactive or active UC group compared to the control group.

Recent studies have shown that feces of IBD patients contain increased amounts of immunoglobulin-coated bacteria and proposed that these Ig-coated bacteria contribute to intestinal inflammation[29–32]. In line with these reports we detected significant elevation of IgA- and IgG-levels in EV fractions from both inactive and active UC patients compared to non-IBD controls (Fig. 2b, c). Neither the literature nor our experience indicate that free Ig can be efficiently precipitated by ultracentrifugation (see IgA-coating of hEVs below), thus the Ig we detected in the EV fractions is apparently bound to pelletable particles and points towards the presence of Ig-EV immunocomplexes. Highest IgA- and IgG-levels were detected in EV fractions from active UC patients and exhibited a positive correlation with the extent of colonic inflammation in UC patients (Fig. 2b, c, Supplementary Fig. 1c, d). SEF of non-IBD and both UC patient groups showed similar IgA-content, which is consistent with previous reports[33]. Thus, UC patients do not exhibit a general increase of intestinal IgA-levels, but may rather be enriched for IgA populations with distinct EV-reactivity resulting in immunocomplex-formation. In contrast, we also detected higher IgG-levels in SEF of UC patients compared to the non-IBD controls (Fig. 2c), suggesting that IgG is generally elevated in the intestinal lumen of UC patients. The increased soluble IgG also correlated with the extent of colonic inflammation in UC patients (Supplementary Fig. 1e).

We chose to focus on IgA in this study, because (i) median IgA-levels were at least an order of magnitude higher than median IgG-levels, (ii) all samples showed detectable IgA-concentrations, and (iii) IgA, unlike IgG, was differentially enriched EV vs. soluble fractions. The Fc-alpha receptor (FcαRI), also known as CD89, is considered the IgA-receptor primarily responsible for IgA-induced pro-inflammatory cytokine responses[34]. Indeed, a role for CD89 in IBD and especially in UC has been proposed, but the specific CD89 function in UC has remained elusive[35–37]. CD89 is a transmembrane protein that can also be detected in solution, i.e., soluble CD89 (sCD89). Complexes of sCD89-IgA have been associated with other inflammatory diseases including IgA vasculitis and IgA nephropathy where they appear to

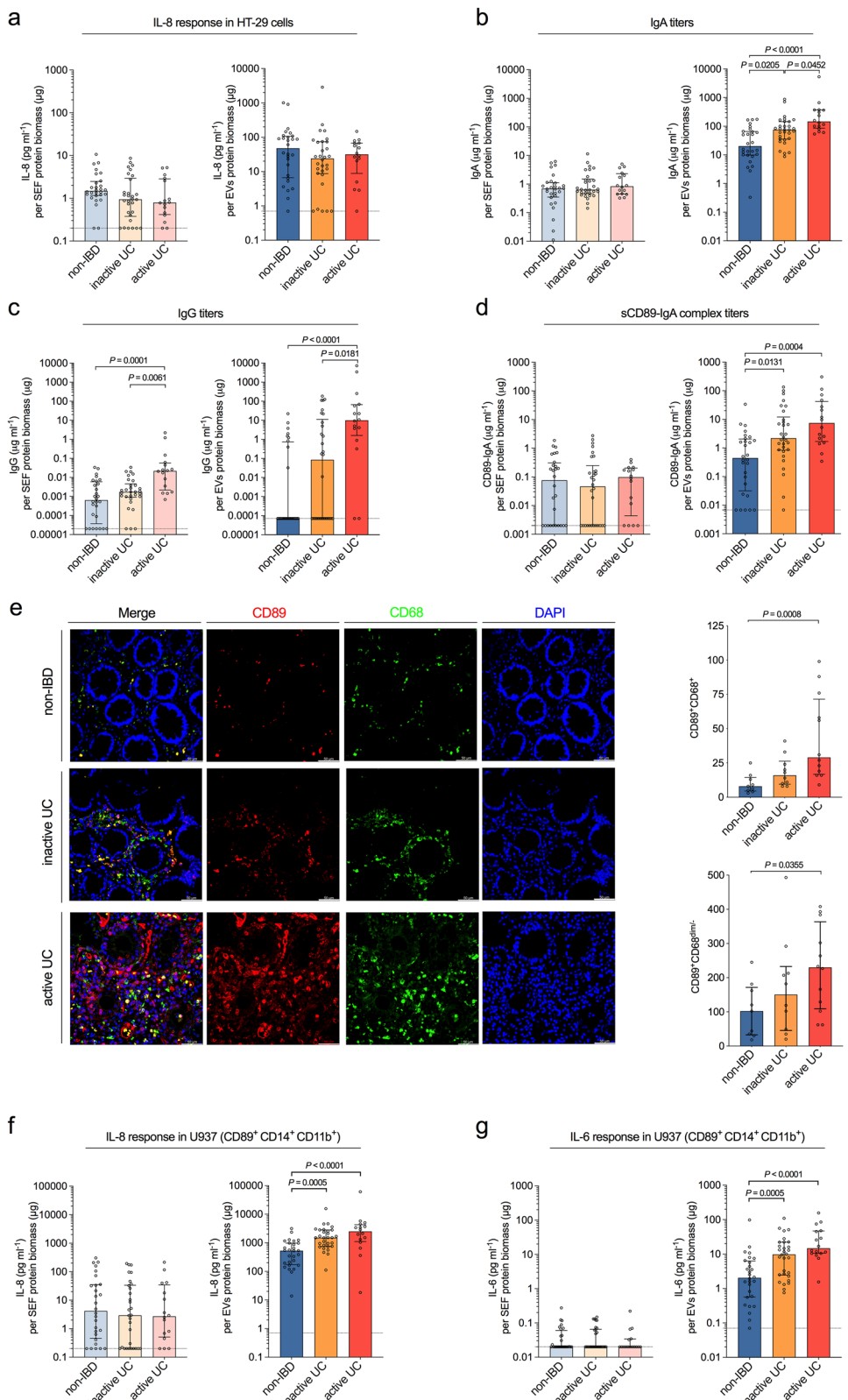

exacerbate inflammation[38–40]. We applied a sandwich ELISA technology combining anti-CD89 and anti-human IgA antibodies for coating and detection to quantitate sCD89-IgA complexes in the soluble and EV fractions (Fig. 2d). EV fractions of UC patients, and most particularly the active UC group, showed increased levels of sCD89-IgA complexes compared to the non-IBD control. Moreover, the amount of sCD89-IgA complexes correlated with the extent of colonic inflammation in UC

patients (Supplementary Fig. 1f). In contrast, soluble fractions of non-IBD, inactive, and active UC groups showed only low amounts of sCD89-IgA complexes.

Unlike other members of the mononuclear phagocyte family, intestinal macrophages lack detectable CD89, which explains low abundance of CD89-expressing cells under steady-state conditions in the healthy gut[41]. However, CD89+ neutrophils and monocytes, can be

**Fig. 2 | Identification of a pro-inflammatory property in EV fractions derived from UC patients. a** IL-8 response in HT-29 human intestinal epithelial cells exposed to soluble effector (SEF, left) and extracellular vesicle (EV, right) fractions of the non-IBD (blue), inactive UC (yellow) and active UC (red) group. **b** IgA titers in soluble effector (SEF, left) and extracellular vesicle (EV, right) fractions of the non-IBD (blue), inactive UC (yellow), and active UC (red) group. **c** IgG titers in soluble effector (SEF, left) and extracellular vesicle (EV, right) of the non-IBD (blue), inactive UC (yellow), and active UC (red) group. **d** sCD89-IgA complex levels in soluble effector (SEF, left) and extracellular vesicle (EV, right) fractions of the non-IBD (blue), inactive UC (yellow), and active UC (red) group. **e** Representative images showing immunofluorescent stainings of human colonic biopsies derived from a non-IBD, inactive UC, and active UC patient for CD89 (red), immune cell marker CD68 (green) and total nuclei stained with DAPI (blue). Graph on the right side depicts the number of CD89+CD68+ and CD89+CD68dim/- cells per eye field for

independent biopsies (non-IBD, $n = 9$; inactive UC, $n = 10$; active UC, $n = 12$). Scale bar = 50 μm. Data in graphs is presented as median ± interquartile range. Statistical difference against the non-IBD group was evaluated by Kruskal–Wallis with Dunn's test. **f** IL-8 response in CD89+CD14+CD11b+ U937 monocytes exposed to soluble effector (SEF, left) and extracellular vesicle (EV, right) fractions of the non-IBD (blue), inactive UC (yellow) and active UC (red) group. **g** IL-6 response in the in CD89+CD14+CD11b+ U937 monocytes exposed to soluble effector (SEF, left) and extracellular vesicle (EV, right) of the non-IBD (blue), inactive UC (yellow) and active UC (red) group. **a–d**, **f**, **g** sample number as follows: non-IBD ($n = 28$), inactive UC ($n = 30$) and active UC ($n = 16$). Data is presented as median ± interquartile range. Samples with no detectable signal were set to the limit of detection, which is indicated by a dotted line. Statistical difference between the three groups was evaluated by Kruskal–Wallis with Dunn's test. **a–g** Source data are provided as a Source Data file.

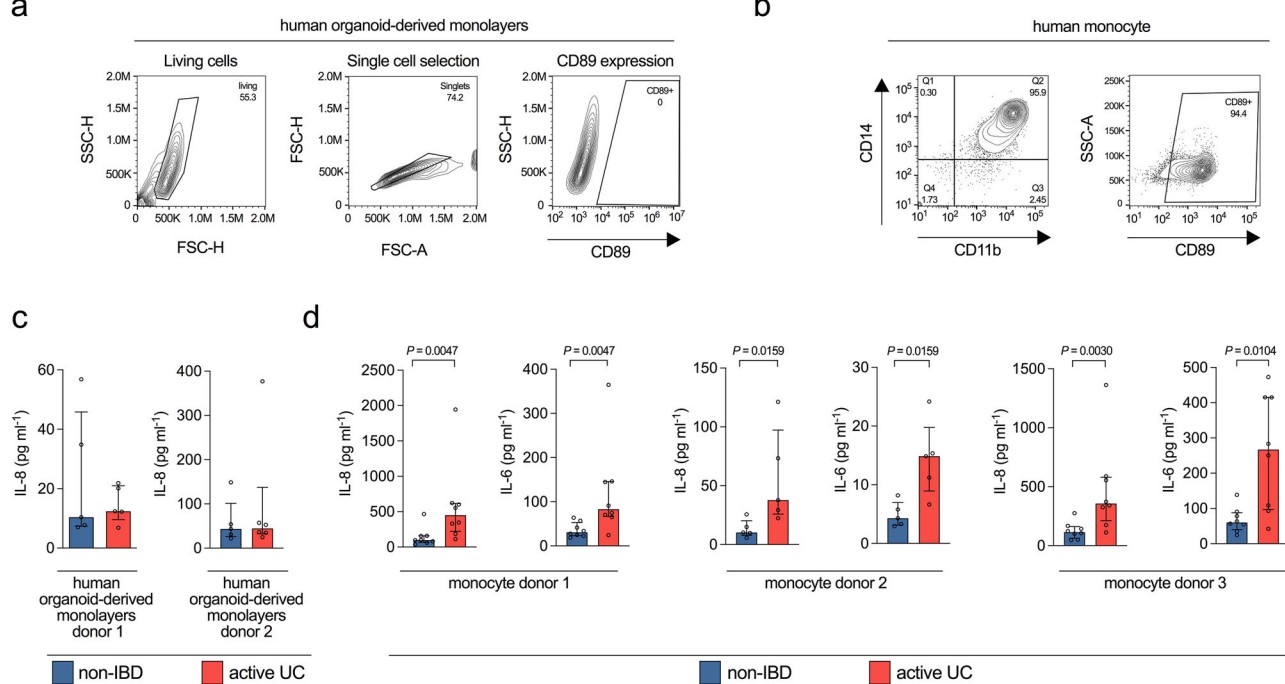

**Fig. 3 | Pro-inflammatory properties of EV fractions derived from UC patients are confirmed in human intestinal organoid-derived monolayers and CD14+ human monocytes. a** Flow cytometry analysis including gating strategy of human intestinal organoids testing for CD89 expression. **b** Flow cytometry analysis of human monocytes isolated from peripheral blood for CD14, CD11b and CD89 expression. **c** IL-8 response in CD89-negative human intestinal organoid-derived monolayers from two healthy donors exposed to extracellular vesicle (EV) fractions derived from the non-IBD (blue) and active UC (red) group (randomly chosen representatives). Lack of CD89 expression was confirmed by FACS provided in

panel a. Data is presented as median ± interquartile range ($n = 6$ for donor 2 exposed to active UC; $n = 5$ for all other data sets). **d** IL-8 (left) and IL-6 (right) response in CD14+ human monocytes isolated from three healthy donors exposed to extracellular vesicle (EV) fractions derived from the non-IBD (blue) and active UC (red) group (randomly chosen representatives). CD89 expression was confirmed by FACS analysis provided in (**b**). Data is presented as median ± interquartile range ($n = 8$ for donor 1 and 3; $n = 5$ for donor 2). **c**, **d** Statistical difference between non-IBD and active UC groups was evaluated by two-sided Mann–Whitney U-test. Source data are provided as a Source Data file.

recruited to the intestinal tissue upon infection or when epithelial barrier function is impaired[42,43]. In agreement with previous reports[44], immunofluorescence imaging of human colonoscopy biopsies showed an increase of CD89+ cells in UC patients (Fig. 2e). Comprehensive quantification revealed significantly increased numbers of CD89+CD68+ cells matching the characteristics of monocyte-derived cells as well as CD89+CD68dim/− cells displaying neutrophilic granulocytes in active UC patients compared to the non-IBD controls. This suggests that abundant CD89+ cells are present in the inflamed tissue of acute UC patients for potential cross-talk with intestinal IgA.

CD89 expression is a unique feature of myeloid cells and therefore epithelial cells, such as HT-29 lack CD89[36,45]. Since we used HT-29 for our initial survey of the pro-inflammatory capacity of colonic fluid samples, we chose next to test monocytic cells, often exhibiting unstable CD89 expression in cell culture. Combined stimulation of the

monocytopoietic cell line U937 with vitamin D3 and TGF-β1 resulted in the induction of CD89+CD14+CD11b+ monocyte-like cells. No stimulation or exposure to only one stimulus resulted in undifferentiated U937 cells, lacking monocyte characteristics with reduced CD89 (Supplementary Fig. 4a). Notably, U937 cells stimulated with vitamin D3 plus TGF-β1 exhibited a similar CD14, CD11b and CD89 expression profile as observed for freshly isolated human peripheral blood monocytes (Fig. 3b and Supplementary Fig. 4a). Although SEF of all three groups failed to induce substantial IL-6 and IL-8 responses in CD89+CD14+CD11b+ U937 monocytes, EV fractions induced robust cytokine responses with significantly increased IL-6 and IL-8 levels for samples from UC patients compared to the non-IBD controls (Fig. 2f, g). Notably, EV fractions derived from the active UC patients, which were shown to contain the highest IgA-levels (Fig. 2b), were also the most inflammatory. In comparison, undifferentiated U937 failed to

elicit a strong pro-inflammatory response with the majority of the values close to or even below the limit of detection (Supplementary Fig. 4b). Purified bacterial peptidoglycan (PGN), which is recognized by toll-like receptors[46,47], significantly induced IL-8 levels in CD89+CD14+CD11b+ U937 monocytes as well as undifferentiated U937 cells (Supplementary Fig. 4c). Although IL-8 levels in undifferentiated U937 were lower, this result demonstrates that undifferentiated U937 are in principle capable of triggering a cytokine response upon exposure to bacterial effectors.

We confirmed the results obtained with CD89-negative HT-29 and CD89+CD14+CD11b+ U937 monocytes, in more physiologically relevant cell systems, i.e., human intestinal organoid-derived monolayers and primary human monocytes (Fig. 3). The CD89 expression profiles analyzed by FACS verified that human intestinal organoid-derived monolayers are CD89-negative, while primary human monocytes are CD89+ (Fig. 3a, b). Unlike HT-29 and U937, which can be continuously passaged and are therefore optimally suited for screens, the primary cell quantity and quality varies between donors. Thus, we randomly selected five to eight representative EV fractions from the active UC patients and non-IBD controls for these assays. EV fractions of active UC patients and non-IBD controls induced similar IL-8 responses in both batches of human intestinal organoid-derived monolayers tested (Fig. 3c). In contrast, EV fractions from UC patients consistently induced significantly higher levels of IL-6 and IL-8 in primary human monocytes from three different donors compared to EV fractions from non-IBD controls (Fig. 3d).

In summary, EV fractions of active UC patients elicit a significantly higher pro-inflammatory cytokine response in CD89 + CD14 + CD11b + U937 monocytes as well as in CD89+ primary human monocytes than EV fractions from non-IBD controls

## IgA-decoration of BEVs heightens pro-inflammatory responses in CD89+ immune cells

Although BEVs are a major constituent of the EV fraction, its composition is complex. The aforementioned host cell cytokine responses to EV fractions (Fig. 2a, f, g) also apply after normalization to the nanoparticle amount (Supplementary Fig. 5), suggesting an immunomodulatory contribution of its major component, i.e., EVs. We applied several independent methods to confirm the presence of bacterial vesicles decorated with IgA in human EV fractions. Immunogold co-staining targeting human IgA (20 nm gold particles) and the bacterial vesicle biomarker LTA or OmpA (6 nm gold particles) visualized the presence of gold particles of both sizes on vesicles in EV fractions (Fig. 4a). We used fluorochrome-conjugated anti-human IgA antibodies combined with nanoparticle tracking analysis (NTA) to quantify the amount of IgA-coated spherical particles (Fig. 4b). Immunoprecipitation using anti-human IgA-coated beads resulted in an enrichment of BEV biomarkers (Fig. 4c, d). In detail, NTA combined with fluorochrome-conjugated anti-LTA antibodies showed a significant enrichment of nanoparticles containing the Gram-positive vesicle biomarker LTA in the pulldown samples compared to original EV fractions (Fig. 4c). Quantitative proteomics by mass spectrometry revealed an enrichment of bacterial vesicle biomarkers in the pulldown samples compared to original EV fractions (Fig. 4d). So far, the formation of IgA-BEV immunocomplexes has been neglected as studies have only focused on bacteria isolated from feces or content of the intestinal lumen without further separation into soluble and pelletable fractions. We applied a sandwich ELISA technology combining anti-LTA (Gram-positive vesicle marker) or anti-OmpA (Gram-negative vesicle marker) as coating antibodies as well as HRP-conjugated anti-human IgA antibodies as detection antibodies to identify BEVs coated with IgA in patient samples. This approach detects only a sub-population of the BEVs, as the LTA- and OmpA-antibody reactivities neither covers all species nor do all BEVs harbor these components. Accordingly, we chose to compare the sample sets from the non-IBD and active UC groups, as these show the biggest difference in IgA-levels

and pro-inflammatory potency. We found significantly higher amounts of both the IgA-LTA and IgA-OmpA complexes in EV fractions from active UC patients compared to non-IBD controls (Fig. 4e). Notably, several samples of the non-IBD group were below the limit of detection. Although absolute quantification is not possible due to the limitations noted above, these data indicate elevated levels of IgA-BEV complexes are present in active UC patients relative to non-IBD patients.

We next asked whether other components present in the complex EV fraction, in particular hEVs, might contribute to the proinflammatory potency observed with CD89+CD14+CD11b+ U937 monocytes. Clear separation of host-derived and BEVs is not possible with current techniques. Further purification steps based on size-exclusion chromatography or density-gradient centrifugation can improve sample purity only marginally and reduces yield[28]. Therefore, we chose to analyze the pro-inflammatory potency of each of the dominant components in the EV fraction individually. We isolated BEVs from lab-cultivated Gram-positive and -negative species (i.e., L. acidophilus, B. fragilis, and ETEC) and compared them to hEVs generated from HT-29 cells. Individual preparations were characterized by standard parameters ensuring reproducible quality and quantity (Table 1). L. acidophilus and B. fragilis were chosen as (i) they represent highly abundant Gram-positive and -negative phyla present in the human gut microbiome[48,49], (ii) they can be cultured in liquid media with sufficient BEV-production, (iii) they are non-flagellated species excluding effects by flagellar components in the BEV preparations and (iv) a recent study reported that BEVs of these species elicited low inflammatory responses in intestinal cells[50]. In contrast, ETEC BEVs originate from a flagellated, pathogenic bacterium and were among the most pro-inflammatory BEVs characterized[50]. Isolated BEVs and hEVs were incubated with commercially available human IgA to allow complex formation. After washing and ultracentrifugation steps to remove unbound IgA the coating was quantified by anti-IgA ELISA (Table 1). Reconstitution of the IgA-coating on BEVs resembled that observed for EV fractions of UC patients. Immunofluorescent and immunogold labeling confirmed the presence of IgA-coated BEVs in the IgA exposed samples, but not in the uncoated control BEVs (Fig. 4a, b). Application of the same decoration protocol for HT-29-derived hEVs resulted in almost no detectable IgA-binding (Table 1). This result is not surprising as substantial binding of human-derived IgA to hEVs originating from human intestinal cells would suggest substantial presence of autoantibodies directed against host epitopes.

Given that patient samples contained a mixture of Gram-positive and -negative BEVs, we mixed IgA-coated and uncoated BEVs from L. acidophilus and B. fragilis (BEVs^B/L) and applied these in varying doses to CD89+CD14+CD11b+ U937 monocytes. Uncoated BEVs^B/L elicited a basic, dose-independent IL-8 and IL-6 response. Conversely, IgA-coated BEVs^B/L induced a dose-dependent cytokine response, which was significantly higher than that induced by uncoated BEVs^B/L for each dose tested (Fig. 5a, b). Similar results were obtained by exposing the individual IgA-coated and uncoated BEVs from L. acidophilus or B. fragilis (BEVs^B and BEVs^L) to CD89+CD14+CD11b+ U937 monocytes (Supplementary Fig. 4d) suggesting that IgA-coated BEVs from both species contribute to the cytokine response. The dose-independent cytokine response observed upon exposure to uncoated BEVs^B/L might indicate that saturation was reached with the BEV amounts tested. Notably, current reports highlight the possibility that interaction of IgA-immunocomplexes with CD89 stimulates phagocytosis[34]. Thus, IgA-coated BEVs may utilize alternative routes, e.g., CD89-dependent pathways, and therefore would be internalized at higher rates and amounts allowing the detection of dose-dependent cytokine responses.

Neither IgA-coated nor uncoated hEVs induced a cytokine response above the mock-treated control. Similarly, soluble human IgA failed to activate an inflammatory response above mock-treatment. Consistent with those findings, CD89 has been reported to bind soluble monomeric

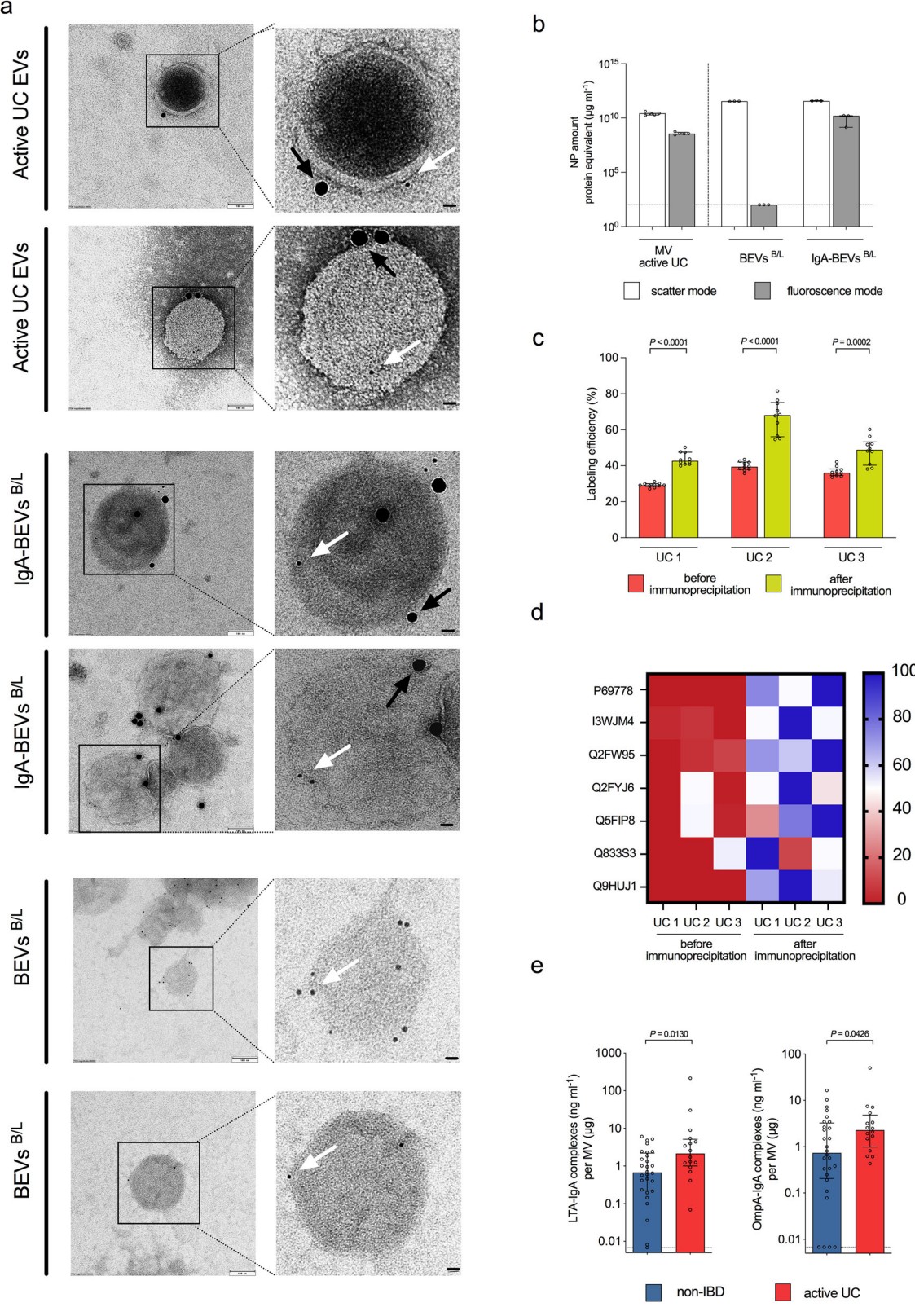

and dimeric IgA with moderate affinity and rather inhibits pro-inflammatory cellular responses[35,36]. In contrast, IgA-immunocomplexes bind avidly to CD89 resulting in CD89 cross-linkage dependent and Syk-mediated pro-inflammatory signaling[35,36]. IgA-coated ETEC BEVs also induced significantly higher IL-8 and IL-6 responses in CD89+CD14+CD11b+ U937 monocytes compared to uncoated ETEC BEVs (Supplementary Fig. 4e). Thus, the stronger pro-inflammatory effect mediated by IgA-coating also holds true for BEVs with inherently high pro-inflammatory capacity. All BEV samples exposed to undifferentiated U937 cells induced only low cytokine responses with no significant differences between IgA-coated and uncoated BEVs for a given dose (Supplementary Fig. 4e–g). IL-8 response evoked by BEVs in HT-29 intestinal epithelial cells, which do not express CD89[36,45], was also independent of IgA-coating (Supplementary Fig. 4h).

**Fig. 4 | Detection of IgA-coated BEVs in human EV fractions. a** Shown are electron micrographs of EV fractions derived from UC patients (randomly chosen representatives), IgA-coated BEVs from *B. fragilis* and *L. acidophilus* (IgA-BEVs[B/L]) and uncoated BEVs from *B. fragilis* and *L. acidophilus* (BEVs[B/L]) after immunogold co-staining for human IgA (20 nm gold particles, black arrow) and the bacterial vesicle biomarkers LTA and OmpA (6 nm gold particles, white arrow). Representative examples of EVs from active UV patients and IgA-BEVs[B/L] show presence of both immunogold particles on the vesicles. Absence of immunogold particles in uncoated BEVs from *B. fragilis* and *L. acidophilus* (BEVs[B/L]) excludes cross-reactivity of the immunogold-conjugated anti-human IgA antibody. The experiments were repeated at least 3 times with a similar result. Scale bars, 100 nm for overview images and 10 nm for the magnified sections. **b** Summarized results from the nanoparticle tracking analysis (NTA) of EV fractions derived from active UC patients (five randomly chosen representatives), uncoated and IgA-coated BEVs from *B. fragilis* and *L. acidophilus* (BEV[B/L] and IgA-BEV[B/L]) after labeling with rabbit anti-human IgA antibody and Alexa Fluor Plus 488- conjugated goat anti-rabbit IgG. For each sample nanoparticle amounts were measured in the conventional light scattering and fluorescent mode. Data is presented as median ± interquartile range (active UC EV fractions, *n* = 5; IgA-BEVs[B/L] and BEVs[B/L], *n* = 3). Samples with no detectable signal were set to the limit of detection, which is indicated by a dotted line. **c** Summarized LTA detection results from the nanoparticle tracking analysis (NTA) of EV fractions derived from active UC patients (three randomly chosen representatives UC1, UC2 and UC3) before and after immunoprecipitation. Samples were labeled with anti-LTA primary antibody and Alexa Fluor Plus 488 secondary antibody and subjected to NTA. For each sample nanoparticle amounts were measured in the conventional light scattering and fluorescent mode. Labeling efficiency (%) of each sample was calculated by dividing the fluorescent particle number to total particle number determined by scattering mode multiplied by 100. Results of 10 measurements for each sample are presented as median ± interquartile range (*n* = 10). Statistical difference between before and after immunoprecipitation was evaluated by two-sided Mann–Whitney U-test. **d** Heat map highlighting the enrichment of bacterial proteins (primary Uniprot accession number) in EV fractions derived from active UC patients after anti-human IgA immunoprecipitation. EV fractions derived from three randomly chosen active UC patients (UC1, UC2, and UC3) were subjected to an immunoprecipitation using an anti-human IgA antibody. Original EV fractions and immunoprecipitated samples were subjected to proteomic analyses. A few surface proteins covering were selected as BEV markers and analyzed as detailed in the method section. Relative abundance in percentile of each protein was assigned with a color code as indicated in the bar on the right side. **e** LTA-IgA (left) and OmpA-IgA (right) effector complex levels in extracellular vesicle (EV) fractions of the non-IBD (blue, *n* = 28) and active UC (red, *n* = 16) group. In the ELISA strategy, anti-LTA or anti-OmpA were used as coating antibodies to capture BEVs in the EV fractions, followed by HRP-conjugated anti-human IgA as detection antibody to quantify the IgA bound to BEVs. Data is presented as median ± interquartile range. Samples with no detectable signal were set to the limit of detection, which is indicated by a dotted line. Statistical difference between non-IBD and active UC groups was evaluated by two-sided Mann–Whitney U-test. **b**–**e**, Source data are provided as a Source Data file.

---

Similar to the EV fraction experiments, we also exposed uncoated BEVs[B/L] and IgA-coated BEVs[B/L] to CD89-negative human intestinal organoid-derived monolayers as well as CD89[+] primary human monocytes to confirm the cytokine responses in more physiologically relevant cell systems (Fig. 5c, d). In line with previous results, primary IgA-coated BEVs[B/L] induced significantly higher pro-inflammatory cytokine responses in human monocytes from three different donors compared to uncoated BEVs[B/L], but not in human intestinal organoid-derived monolayers.

In summary, IgA-BEV immunocomplexes can drive a cytokine release in CD89[+]CD14[+]CD11b[+] U937 monocytes as well as CD89[+] primary human monocytes exacerbating inflammatory responses.

## IgA-coated BEVs exacerbate DSS-induced colitis in human CD89-expressing mice

We next sought to address the role of IgA-BEV immunocomplexes in UC using an in vivo system. As mice do not have a CD89 homolog, we used a transgenic (Tg) mouse line expressing human CD89 within the myeloid lineage[51]. Earlier work confirmed functionality of the CD89 downstream cascades in this mouse line including phosphorylation of Syk and Erk[51]. The design of the CD89 Tg mouse line generates CD89[tg/wt] (CD89-transgenic mice) or CD89[wt/wt] littermates with or without the CD89 expression allowing side-by-side comparison of age-matched cohorts[51]. Presence of CD89[+] cells in CD89-transgenic mice, but not in littermates was confirmed by FACS of bone-marrow derived cells (BMDCs) (Supplementary Fig. 6a). We then exposed BMDCs isolated from littermates and CD89-transgenic mice to varying doses of IgA-coated and uncoated BEVs[B/L] to verify that IgA-BEVs immunocomplexes also exacerbate inflammatory responses in mouse-derived CD89[+] cells in a dose-dependent manner (Fig. 6a). In line with results obtained for the human U937 cell line as well as primary human monocytes, highest cytokine responses were observed in CD89[+] BMDCs in response to IgA-coated BEVs[B/L] for each dose tested. The IL-6 and TNF-α levels of CD89[+] BMDCs exposed to IgA-coated BEVs[B/L] were significantly increased compared to CD89[+] BMDCs exposed to uncoated BEVs[B/L] or to littermate BMDC exposed to IgA-coated or uncoated BEVs[B/L] for two or more BEV doses tested. As demonstrated recently, the tyrosine kinase Syk represents an essential downstream factor in the CD89-dependent signaling pathway in CD89-transgenic mice[51]. Treatment of CD89[+] BMDCs with the Syk-inhibitor R406 significantly decreased IgA-coated BEVs[B/L]

mediated IL-6 and TNF-alpha responses (Fig. 6b). BMDC isolated from littermates and CD89-transgenic mice are ideally suited to demonstrate CD89-dependent effects as they only differ in the expression of human CD89. Thus, we also assessed their cytokine responses upon exposure to seven randomly selected representative EV fractions from active UC patients and non-IBD controls (Fig. 6c). In BMDCs isolated from littermates the EV fractions from both groups, i.e., active UC patients and non-IBD controls, induced similar, low level cytokine responses. Consistent with observations made with the CD89[+]CD14[+]CD11b[+] U937 monocytes and primary human monocytes, EV fractions from active UC patients induced significantly higher IL-6 and TNF-alpha responses in CD89[+] BMDCs compared to non-IBD controls. Treatment of CD89[+] BMDCs with the Syk-inhibitor R406 significantly decreased IL-6 and TNF-alpha responses mediated by EV fractions from three randomly selected active UC patients (Fig. 6d). These results underpin the importance of CD89 including its downstream pathways for the strong pro-inflammatory responses evoked by IgA-coated BEVs[B/L] as well as by the EV fractions from active UC patients.

Mild colitis was induced in CD89-transgenic mice and littermates by dextran sodium sulfate (DSS) for 4 days as indicated by moderate histological colitis and low levels of intestinal lipocalin, serving as a biomarker for intestinal inflammation[52–54] (Supplementary Fig. 6b, c). Importantly, CD89-transgenic mice and littermates showed comparable signs of inflammation at this time point suggesting similar susceptibility to DSS. To test the impact of BEVs, mice with DSS-induced mild colitis were orally gavaged twice a day for five days with IgA-coated BEVs[B/L], uncoated BEVs[B/L] or the solvent control (saline) (Fig. 7a). Human IgA was readily detectable in the cecal content of mice receiving IgA-coated BEVs[B/L], but not on mice receiving uncoated BEVs[B/L] (Supplementary Fig. 6d). Thus, the effectors applied by oral gavage have reached the relevant areas in the gut. Only CD89-transgenic mice receiving IgA-coated BEVs[B/L] exhibited severe signs of inflammation highlighted by a significant shortening of the colon, significantly elevated levels of intestinal lipocalin, and significantly increased colitis scores in histology readouts (Fig. 7b–d). Neither treatment of CD89-transgenic mice with uncoated BEVs[B/L] or saline nor any treatment group of the littermates showed such an effect. Direct histological comparison of the mice gavaged with IgA-coated BEVs[B/L] revealed that CD89-transgenic mice showed increased neutrophil infiltration and extensive loss of tissue integrity, which was neither

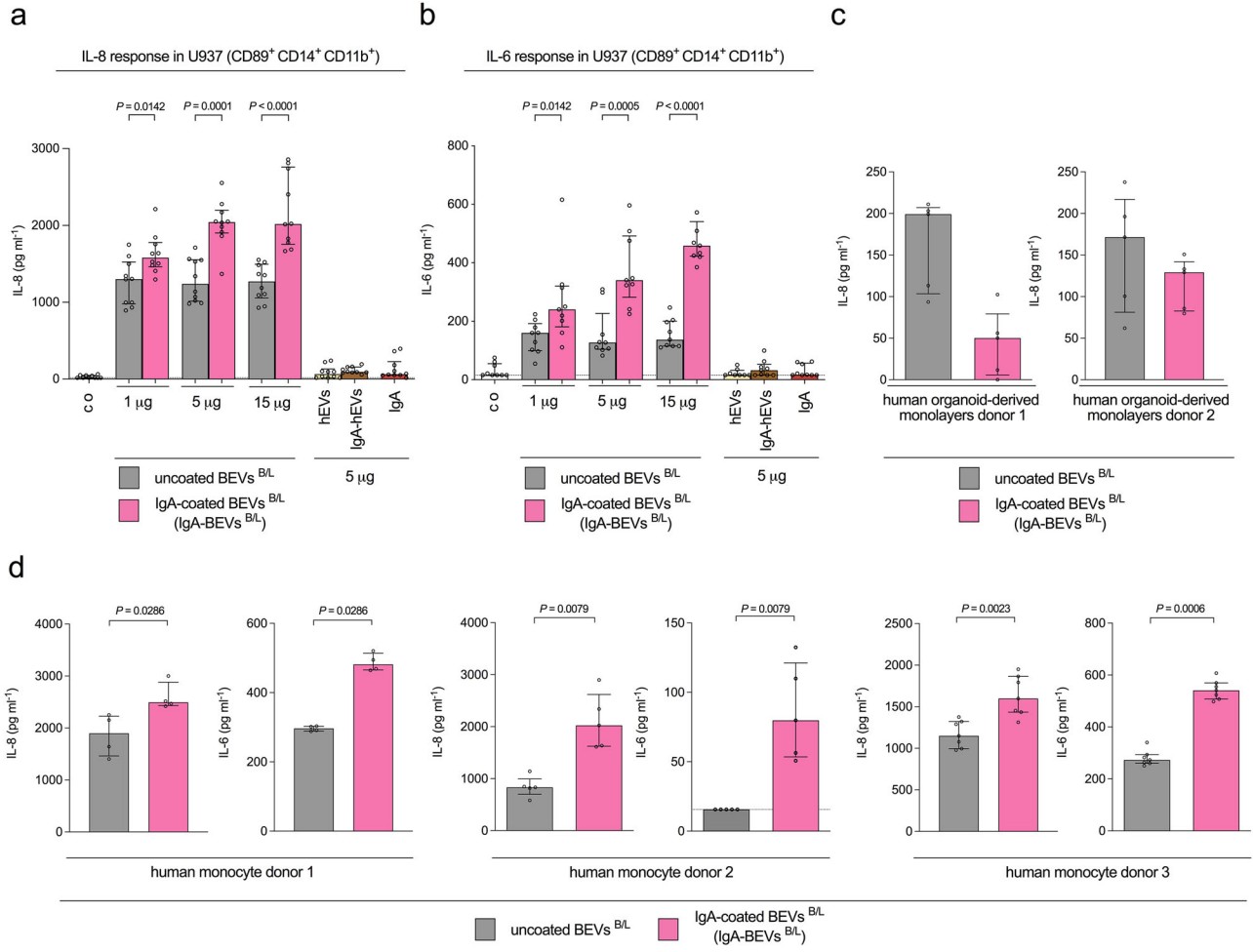

**Fig. 5 | IgA-coated BEVs promote pro-inflammatory cytokine release in CD89⁺ cells. a** IL-8 response in CD89⁺CD14⁺CD11b⁺ U937 monocytes exposed to uncoated or IgA-coated BEVs from *B. fragilis* and *L. acidophilus* (BEVs^B/L and IgA-BEVs^B/L), uncoated or IgA-coated hEVs or soluble IgA alone. Dosage of the effector samples given by total protein biomass determined by Bradford is indicated. Incubation with saline served as mock-treated control (co). (*n* = 10 for each data set). **b** IL-6 response in CD89⁺CD14⁺CD11b⁺ U937 monocytes exposed to uncoated or IgA-coated BEVs from *B. fragilis* and *L. acidophilus* (BEVs^B/L and IgA-BEVs^B/L), uncoated or IgA-coated hEVs or soluble IgA alone. Dosage of the effector samples given a total protein biomass determined Bradford is indicated. Incubation with saline served as mock-treated control (co). (*n* = 8 for uncoated hEVs; all other data sets *n* = 9). **c** IL-8 response in CD89-negative human intestinal organoid-derived

monolayers from two healthy donors exposed to 5 µg uncoated or IgA-coated BEVs from *B. fragilis* and *L. acidophilus* (BEVs^B/L and IgA-BEVs^B/L). Lack of CD89 expression was confirmed by FACS (Fig. 3a). Data is presented as median ± interquartile range (*n* = 5 for each data set). **d** IL-8 (left) and IL-6 (right) response in CD14⁺ human monocytes isolated from three healthy donors exposed to 5 µg uncoated or IgA-coated BEVs from *B. fragilis* and *L. acidophilus* (BEVs^B/L and IgA-BEVs^B/L). CD89 expression was confirmed by FACS analysis (Fig. 3b). Data is presented as median ± interquartile range (*n* = 4 for donor 1, *n* = 5 for donor 2, and *n* = 7 for donor 3). **a–d** Data is presented as median ± interquartile range. Samples with no detectable signal were set to the limit of detection, which is indicated by a dotted line. Statistical difference between uncoated and IgA-coated BEVs was evaluated by two-sided Mann–Whitney U-test. Source data are provided as a Source Data file.

observed in littermates receiving the same treatment nor any mice treated with uncoated BEVs or saline (Fig. 7e and Supplementary Fig. 6e). Immunofluorescent imaging of colon tissue demonstrated substantial presence of CD89⁺ cells in CD89-transgenic mice receiving IgA-coated BEVs^B/L, but not in littermates receiving the same treatment (Fig. 7f). These in vivo data indicate that IgA-coated BEVs are sufficient to drive intestinal inflammation in a CD89-dependent manner.

## Discussion

Current models propose that the intestinal microbiome as well as an aberrant immune response are likely to be important triggers for the prolonged periods of intestinal inflammation observed in UC. Here we report that intestinal levels of IgA-coated BEVs are elevated in UC patients and further demonstrate that physiological concentrations of these effectors are potent activators of pro-inflammatory cytokine release in CD89⁺ immune cells. Moreover, IgA-coated BEVs exacerbate colitis in a CD89-transgenic mouse model of UC. Our data provide

functional evidence linking microbial-derived effectors, i.e., BEVs, with host-derived factors, i.e., immunoglobulins and immune cells, that together contribute a molecular explanation for the inflammatory condition observed in UC pathogenesis.

Previous reports showed that levels of IgA-opsonized of intestinal bacteria are elevated in IBD patients compared to healthy controls, but the relevance of this observation to the disease has not been clarified[29–32,55–57]. Here we demonstrated that a pronounced UC-related immunoglobulin-coating is mirrored in the BEVs present in these patients. Given the species-specific composition and cargo of BEVs, different populations exhibit distinct immunomodulatory properties and mediate discrete host cell receptor recognition[28,50]. However, our results based on HT-29 cell responses highlight that the EV fractions of UC patients are per se not particularly pro-inflammatory. In contrast, our findings identify IgA-coating as key prerequisite for efficient recognition of BEVs by CD89⁺ cells which respond with release of pro-inflammatory cytokines. Importantly, formation of BEVs is activated by

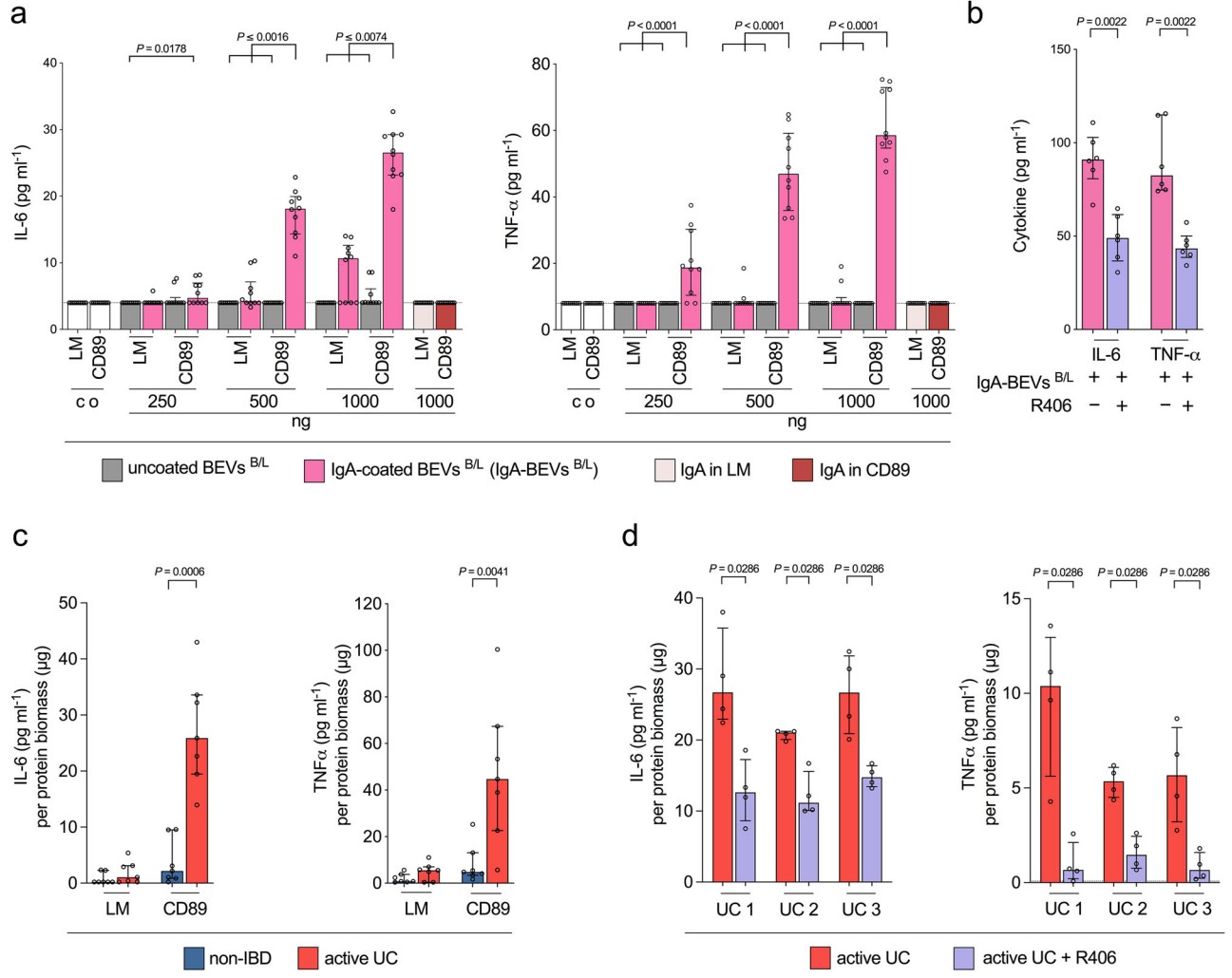

**Fig. 6 | IgA-coated BEVs and EV fractions from UC patients promote a CD89-dependent pro-inflammatory cytokine release in mouse bone marrow derived cells (BMDC). a** IL-6 and TNF-α response in bone marrow derived cells (BMDC) from CD89[wt/wt] (littermates, LM) and CD89[tg/wt] (CD89-transgenic, CD89) mice exposed to uncoated or IgA-coated BEVs from *B. fragilis* and *L. acidophilus* (BEVs[B/L] and IgA-BEVs[B/L]) as well as soluble IgA alone. Dosage of the effector samples given by total protein biomass determined by Bradford is indicated. Incubation with saline served as mock-treated control (co). (*n* = 10 for each data set). **b** IL-6 and TNF-α response in bone marrow derived cells (BMDC) from CD89[tg/wt] (CD89-transgenic, CD89) mice exposed to 1 μg IgA-coated BEVs from *B. fragilis* and *L. acidophilus* (IgA-BEVs[B/L]) in the absence (-) or presence (+) of the Syk inhibitor R406. (*n* = 6 for each data set). **c** IL-6 (left) and TNF-α (right) response in bone marrow derived cells (BMDC) from CD89[wt/wt] (littermates, LM) and CD89[tg/wt] (CD89-transgenic, CD89)

mice exposed to EV fractions derived from the non-IBD (blue) and active UC (red) group (seven randomly chosen representatives). (*n* = 7 for each data set). **d** IL-6 and TNF-α response in bone marrow derived cells (BMDC) from CD89[tg/wt] (CD89-transgenic, CD89) mice exposed to 1 μg EV fractions derived from three representative active UC patients (UC1, UC2, or UC3) in the absence (-) or presence (+) of the Syk inhibitor R406. (*n* = 4 for each data set). **a**–**d** Data is presented as median ± interquartile range. Samples with no detectable signal were set to the limit of detection, which is indicated by a dotted line. Statistical difference in panel a was evaluated by Kruskal–Wallis (comparison of groups receiving the same BEVs[B/L] dosage) with Dunn's test. Statistical differences in all other panels [comparison of (B)EV samples with and without inhibitor or non-IBD and active UC groups] were evaluated by two-sided Mann–Whitney U-test. Source data are provided as a Source Data file.

stressors present in the human gut, thus these effectors may be highly abundant in the intestinal tract[15,58]. Representing non-living facsimiles of the donor bacterium, BEVs are ideally suited for host cell-interaction studies. Thus, the protocol presented here for EV isolation from colonic fluid might be a valuable tool for future research targeting other intestinal diseases besides UC.

Given the complexity of UC, it is likely that additional factors and pathways can also fuel the chronic inflammation. For example, we also observed elevated IgG-levels in colonic luminal fractions of UC patients. Although IgG-levels were at least an order of magnitude lower compared to IgA-titers, increased IgG-titers may also drive chronic inflammation and TH17 immunity as previously reported[32].

CD89[+] immune cells have been understudied in IBD research since human intestinal macrophages do not express CD89 and wildtype

mouse models used to recapitulate IBD pathogenesis lack the CD89 receptor[41,51]. We also observed very few CD89[+] cells in biopsies of non-IBD patients. While this is apparently true during homeostasis in a healthy gut, here we demonstrate massive infiltration of CD89[+] cells in UC patients, particularly during active colitis. Since impaired intestinal epithelial barrier function is a hallmark of IBD, it is likely that leakage of luminal content into the mucosa recruits immune cells from systemic circulation.

Here and elsewhere, CD89 was not activated by soluble IgA, but rather requires IgA-immunocomplexes, for example by IgA-opsonized bacteria as earlier data suggest[35,36]. In this study we demonstrate that IgA-coated BEVs, representing non-living facsimiles of the bacteria, can act as immunocomplexes potently activating immune cells in a CD89-dependent manner. Although IgA-immunocomplexes can directly

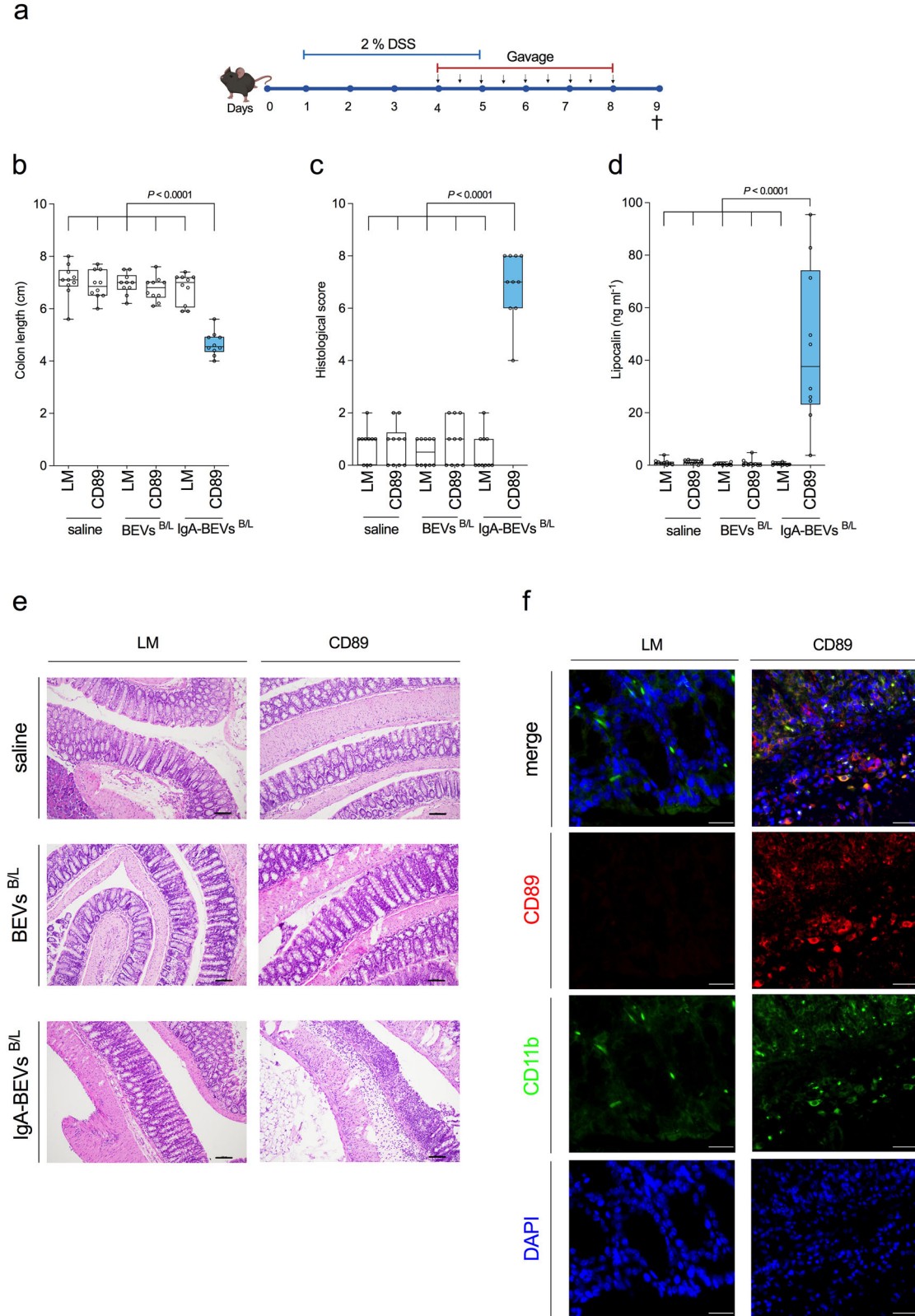

activate CD89-dependent functions such as phagocytosis and degranulation[34], this interaction is only partly sufficient to induce cytokine production. To this end, CD89 activates an ITAMs-dependent downstream cascade and collaborates with other receptors, such as PRR, which have been characterized as effective sensors for various BEV compounds resulting in pro-inflammatory cytokine release[34,35].

In summary our data suggest a direct link of IgA-coated BEVs and CD89⁺ immune cells in UC. Based on our results, we propose that UC patients harbor high levels of IgA-coated BEVs, which act as immuno-complexes activating CD89⁺ cells. Under homeostatic conditions IgA-opsonized bacteria are primarily restricted to the intestinal lumen, while under pathological conditions they can reach the lamina propria

**Fig. 7 | IgA-coated BEVs exacerbate DSS-induced colitis in CD89+ mice.**
**a** Timeline of the DSS treatment of CD89^wt/wt (littermates, LM) and CD89^tg/wt (CD89-transgenic, CD89) mice followed by oral gavage with uncoated or IgA-coated BEVs from *B. fragilis* and *L. acidophilus* (BEVs^B/L and IgA- BEVs^B/L). Oral gavage with saline served as mock-treated control (co). Created in BioRender. Fleischhacker, D. (2025) https://BioRender.com/hbvosjv. **b** Colon length determined along endpoint harvest (day 9). **c** Colon histopathology scores assessed double-blinded from colon harvested at the endpoint (day 9). Scores are the sum of the individual criteria assessment including epithelial alteration, neutrophil infiltration and erosion assigned as follows: 0, absent; 1, focal; 2, present; 3, abundant. **d** Lipocalin-2 levels determined by ELISA from cecal content harvested at the endpoint (day 9). **b**–**d** Data is presented as box plot showing minimum and maximum (whiskers), interquartile range (bounds of the box), median (center) (*n* = 10 mice/group). Statistical difference was evaluated by Kruskal–Wallis (CD89-transgenic mice receiving IgA-coated BEVs^B/L to all other groups) with Dunn's test. Source data are provided as a Source Data file. **e** Representative histology pictures from heamotoxylin and eosin stained colons derived from CD89^wt/wt (littermates, LM) and CD89^tg/wt (CD89-transgenic, CD89) mice receiving saline, uncoated or IgA-coated BEVs from *B. fragilis* and *L. acidophilus* (BEVs^B/L and IgA- BEVs^B/L) harvested at the endpoint (day 9). Note that CD89-transgenic mice exhibit intensive neutrophil infiltration, epithelial alteration and erosions, whereas littermates showed no or only minimal signs of inflammation. The experiments were repeated at least 10 times with a similar result. Scale bar = 100 μm. **f** Representative images showing immunofluorescent stainings of colon tissue derived from CD89^wt/wt (littermates, LM) and CD89^tg/wt (CD89-transgenic, CD89) mice receiving IgA-coated BEVs^B/L harvested at the endpoint (day 9). Tissue specimen were stained with anti-human CD89 (red), anti-mouse CD11b (green) and total nuclei stained with DAPI (blue). The experiments were repeated at least 3 times with a similar result. Scale bar = 50 μm.

to interact with the immune cells[59,60]. Similar principles may apply to BEVs, but whether IgA-coated bacteria or BEVs are more important in exacerbating inflammation remains to be elucidated. The direct visualization of BEVs and their differentiation from bacterial cells in the human intestine is limited by the available methods, as current light microscopy cannot visualize vesicle with a sufficiently high resolution. However, BEVs are non-living facsimiles of the donor cell vesicles and therefore share the same epitopes as the bacterial cell. Mechanistic studies on how BEVs spread in vivo are limited. However, it is believed that BEVs can pass the intestinal barrier more easily than bacteria and can deliver a concentrated dose of pro-inflammatory cargo, e.g., LPS, LTA, surface proteins or peptidoglycan[15,61]. Thus, we propose a mechanism whereby an initial event of epithelial barrier or mucosal layer dysfunction might be required to allow a first contact. IgA-BEV recognition by CD89 and subsequent CD89-PRR crosstalk would results in pro-inflammatory cytokine release, which in turn would facilitate infiltration of more CD89+ cells, prevent mucosal healing, and sustain continuous leakage of IgA into the intestinal lumen. This would generate more IgA-coated bacteria or BEVs, which then activate newly recruited CD89+ cells. Overall, this scenario creates a viscous cycle of ongoing inflammation as observed in UC. This model is supported by our mouse study where epithelial damage leading to colitis was first induced by DSS. The application of IgA-coated BEVs resulted in an aggravation and perpetuation of colitis in mice expressing the CD89 receptor, while the control group recovered.

Our results have several clinical impacts. BEVs originating from the gut microbiota have been so far fairly neglected as relevant effectors in IBD. Correlation of IgA-coated BEVs and colonic sCD89-IgA levels with the degree of inflammation suggests their potential as suitable UC biomarkers. Current treatment of UC primarily focuses on modulation of innate immune system or T-cell pathways, but half of the patients show no remission and 15% require colectomy[2,3,5,6]. Failure of the anti-CD20 antibody rituximab in a pilot study of active UC dampened enthusiasm for B cell targeted approaches[62]. However, IgA-positive mucosal plasmablasts and plasma cells, which are enriched in active UC[63], do not display CD20 on their cell surface[64]. Thus, the potential efficacy of a B cell targeted approach may have been masked in this study. Furthermore, currently used IBD mouse models lack the CD89 receptor and might therefore fail to recapitulate some inflammatory pathways which are relevant for human UC. Considering the inflammatory potency of IgA-coated BEVs reported here, we propose that adaptive B cell immune processes warrant renewed attention in drug development strategies.

## Methods
### Ethics statement
This study conformed to the principles of the Declaration of Helsinki, Good Clinical Practice Guidelines and the protocol for obtaining human samples was approved by the local ethics committee of the Medical University of Graz, (17–199 ex 05/06, 34-460 ex 21/22 and 36-116 ex 23/24). Written informed consent was obtained from all patients included in the study. Patient samples were pseudonymized by the clinicians after harvest. The study protocol was published under clinicaltrials.gov (NCT04136587). The protocol for the establishment of human organoid cultures was approved by the ethics committee of the Medical University of Vienna (1260/2022).

All mouse studies were performed in accordance with the Commission for Animal Experiments of the Austrian Ministry of Science (animal protocol: GZ BMWFW-39/8/75 ex 2020/21) and the local Animal Welfare Committee of the University of Graz (Head: Kathrin Zierler). Mice were housed in a specific-pathogen-free classified facility in individually ventilated cages (IFC) and maintained on a 12 h light/dark cycle at 21 °C and 48% humidity. For the genetically modified strains littermates (LM) were used as controls. Food and water were offered ad libitum. Mice were monitored daily for stress levels and weight was measured every second day. Mice were killed at the study endpoint with isoflurane and cervical dislocation.

### Patient enrollment, clinical examination, classification into groups
Study subjects undergoing scheduled colonoscopy at the endoscopy unit of the Division of Gastroenterology and Hepatology (Medical University of Graz) were recruited for the study between May 2019 and September 2021. General inclusion criteria were an age > 18 years and a signed informed consent, a general exclusion criterion was pregnancy. Inclusion and exclusion criteria for the particular study groups were as follows:

Non-IBD control group (*n* = 28): Inclusion criteria: negative history for intestinal and autoimmune diseases and colonoscopy performed for the following indications: anemia, blood in stool, constipation, change in bowel habits, screening for colon cancer, follow up after colonic polyps and weight loss; a macroscopic normal colonoscopy except for diverticulosis (without any signs of inflammation), ≤3 small polyps (except of hyperplastic polyps of the colon and rectum) or angiodysplasia. Exclusion criteria: Diagnosis of IBD or any other inflammatory condition of the small and large intestine, diagnosis of irritable bowel syndrome, acute or chronic diarrhea, autoimmune disorders, obesity (body mass index ≥30), regular intake of non-steroidal anti-inflammatory drugs (NSAIDs; >2 tablets/ week) or immunosuppressants, intake of antibiotics within the last 3 months, intestinal infection by enteric pathogens and probiotic therapy;

UC group (*n* = 46): Inclusion criteria: established or suspected (which was later confirmed) diagnosis of UC by based on the current guidelines of the European Crohn's and Colitis Organization[65]; Exclusion criteria: Diagnosis of Crohn´s disease and infections by enteric pathogens;

At the time of endoscopy demographic data, medical history, gastrointestinal symptoms and intake of medications were assessed from the patients. To evaluate clinical activity of UC, the total Mayo

score was recorded[66,67]. The Mayo endoscopic subscore was assessed by endoscopists experienced in IBD and confirmed by an independent blinded endoscopist from electronic images of the site of most severe inflammation[61,67,68]. In addition, the quality of the bowel preparation was scored using the validated OBPQS, as this score includes also the residual fluid in the colonic lumen[17]. Furthermore, the maximal extent of colonic inflammation in UC patients was assessed according to the Montreal classification, which distinguishes between proctitis (1), left sided colitis (2) or extensive colitis and pancolitis (3)[69]. UC patients were classified into two groups according to the inflammation seen on colonoscopy in inactive UC was defined by a Mayo endoscopic subscore of 0 or 1 ($n = 30$) and active UC as a Mayo endoscopic subscore of 2 or 3 ($n = 16$)[66–68].

Using the same assessment of UC activity as well as the same inclusion and exclusion criteria (except the number of polyps in non-IBD patients), human biopsies for the immunofluorescence staining were collected from independent patients during 2023 and 2024 (Supplementary Table 1). Eight forceps biopsies from the sigmoid colon at 30 cm were obtained and put into a tube containing PBS solution. The biopsies were cooled on crushed ice and immediately transferred to the immunology laboratory.

### Collection of colonic luminal fluid and separation into soluble and pelletable (EV) effector fractions

The evening before colonoscopy patients started standard bowel preparation for colonoscopy using a PEG/ascorbate-based solution (Moviprep or Pleinvue Norgine, Austria, PZN 3778668) with a split dosing regimen. Colonoscopy was performed under standard sedation. During colonoscopy colonic content was collected by suction of the residual fluid and stool contents of the colonic lumen and also by suction of washing fluid (sterile $H_2O$ solution without additives, Ampuwa, Fresenius Kabi Austria, PZN 2973853). During colonoscopy all three main segments of the colon (right colon; transverse colon, left colon/rectum) were washed continuously with $H_2O$ to ensure that also microbial contents adhering to the bowel wall were collected. Additional washing was performed as required for clear view of the mucosa for endoscopy. The amount of $H_2O$ used for washing of the muscosa was recorded after colonoscopy. For every patient and colonoscopic procedure a new tube system and collector system was used. After colonoscopy the container with the aspirated colonic luminal content was placed on crushed ice and transferred to the laboratory at the University of Graz in an enclosed ice box within 1 h.

Colonic luminal samples were centrifuged (10,000 $g$, 4 °C, 20 min) and subsequently sterile-filtered through a 0.22 μm pore size bottle-top-filter (PES, VWR International) to remove solid material, living cells, and cellular debris. Removal of living bacterial cells was routinely confirmed by plating 1 ml on BHI agar plates following incubation overnight. Filtered supernatant was subjected to ultra-centrifugation (150,000 $g$, 4 °C, 4 h) using a 45Ti rotor separating the SEF (supernatant) from PEF (pellet). Top 50 ml of the supernatant reflecting the SEF were carefully removed. The residual supernatant was discarded. The pellet reflecting the PEF was resuspended in saline to generate a PEF suspension 100-fold concentrated compared to the original colonic luminal sample. SEF and PEF were subjected to further analysis or stored at −80 °C until further use. PEF samples were analyzed for presence of EVs by routine quality and quantity assays in line with the MISEV guidelines for EV samples[19], such as NTA, Bradford assays, transmission electron microscopy, lipid quantification, nucleic acid quantification, LAL assays, LTA ELISA, and presence of CD9, CD81 and CD63 by dot blot.

### Bacterial strains, growth conditions and bacterial extracellular vesicle (BEV) isolation

Bacterial strains used in this study for the isolation of BEVs were *B. fragilis* ATCC 25285, *L. acidophilus* ATCC 4356 and a spontaneous streptomycin-resistant mutant of ETEC H10407[50] (Supplementary Table 2). Bacteria were grown at 37 °C in brain heart infusion (BHI) broth or on BHI agar (Thermo Fisher Scientific, 237300). Streptomycin (Sigma-Aldrich, S6501) was used in the final concentrations of 100 μg ml⁻¹. BEVs were isolated according to previous reports as follows[50]:. Overnight cultures of ETEC were grown with aeration (180 rpm), while overnight cultures of *B. fragilis* and *L. acidophilus* were grown anaerobically using GasPak EZ systems (BD Biosciences, 260672). The respective overnight cultures were diluted (1:100) in BHI medium and grown at 37 °C either with aeration for 8 h in case of ETEC or overnight anaerobically using GasPak EZ systems (BD Biosciences, 260672) in case of *B. fragilis* and *L. acidophilus*. Culture supernatant was cleared from bacterial cells by centrifugation (10,000 $g$, 20 min) and subsequent sterile filtration (0.22-μm pore size bottle top filters). Removal of living bacterial cells was confirmed by plating 1 ml on BHI agar plates following incubation overnight. BEVs present in the sterile-filtered supernatant were pelleted through subsequent ultra-centrifugation (150,000 $g$, 4 °C, 4 h) and resuspended in saline to generate a BEV suspension 1000-fold concentrated compared to the original culture supernatant. BEV samples were subjected to routine quality and quantity controls [NTA, Bradford assays, transmission electron microscopy, lipid quantification, nucleic acid quantification, LAL assays (in case of Gram-negative BEVs) and LTA ELISA (in case of Gram-positive BEVs)] and stored at −20 °C until further use.

### Cell culture conditions

The human intestinal epithelial cell line HT-29 (ATCC, HTB-38)[50], human CD14⁺ monocytes isolated from peripheral blood and the human pro-monocytic myeloid leukaemia cell line U937 (ATCC, CRL-1593.2)[70] were used in this study (Supplementary Table 2). Cell lines were maintained at 37 °C and 5% $CO_2$ in a humidified atmosphere and cultivated in DMEM/F-12 growth medium (Thermo Fisher Scientific, 21331-020) supplemented with 2 mM L- glutamine (Thermo Fisher Scientific, 25030-081) in case of HT-29 or RPMI 1640 growth medium (Sigma-Aldrich, R8758) in case of U937 and CD14⁺ monocytes. Unless stated otherwise growth medium was supplemented with 10% fetal bovine serum (FBS, Thermo Fisher Scientific, 10270-106), 50 μg ml⁻¹ penicillin and 50 μg ml⁻¹ streptomycin (Thermo Fisher Scientific, 15140-122).

### Host-derived vesicle (hEV) purification from HT-29 cells

The hEVs were isolated from the human intestinal epithelial cell line HT-29. In brief, supernatant of confluent HT-29 cells was centrifuged (4000 $g$, 4 °C, 20 min) and subsequently sterile-filtered through a 0.22-μm pore size bottle-top-filter to remove living cells and cellular debris. Vesicles present in the sterile-filtered supernatant were pelleted through ultracentrifugation (150,000 $g$, 4 °C, 4 h) and resuspended in saline to generate a hEV suspension 1000-fold concentrated compared to the original culture supernatant. Isolated hEVs were subjected to routine quality and quantity controls (NTA, Bradford assays, transmission electron microscopy, nucleic acid quantification, lipid quantification and presence of CD9, CD81 and CD63 markers by dot blot) and stored at −20 °C until further use.

### Antibodies and oligonucleotides

Antibodies used in this study are listed in Supplementary Table 3 including their final concentrations or dilutions for the respective assays. Oligonucleotides used in this study are listed in Supplementary Table 4.

### Protein quantification

Protein concentrations of soluble and PEFs derived from human colonic luminal samples, BEVs and hEV preparations were determined by Bradford assays (Bio-Rad Laboratories, Protein Assay Dye Reagent, #5000006) according to the manufacturer's manual and according to

previous reports as follows[50]: To allow detection of luminal content of vesicles samples were lysed with 0.1% SDS (Roth, CN30.3) for 10 min prior to the assay. 0.1% SDS is the highest concentration compatible with the assay[71]. While addition of 0.1% SDS increases the amount of quantified proteins in a variety of EVs derived from diverse Gram-positive and Gram-negative bacteria[50], we cannot exclude insufficient lysis for some EV types present in the samples.

## LPS reactivity

LPS amounts in PEFs derived from human colonic luminal samples and BEVs derived from Gram-negative species were determined by the Pierce Chromogenic Endotoxin Quant kit, according to the manufacturer's instructions (Invitrogen, A39553). Briefly, appropriate dilutions of the samples and LPS standards were adjusted to a volume of 50 µl and incubated with 50 µl limulus amoebocyte lysate for 9 min. Chromogenic substrate solution was added and samples were incubated for 6 min at 37 °C, then the reaction was stopped by the addition of 25% (v/v) acetic acid. Absorbance was measured at 405 nm using a SPECTROstar nano plate reader (BMG Labtech, Germany) and the amount of LPS associated with the samples was quantified using the standard curve (0.1–1.0 EU ml$^{-1}$), according to the manufacturer's instructions. Each assay was performed at least in technical duplicate.

## LTA quantification

Levels of LTA in PEFs derived from human colonic luminal samples and BEVs derived from Gram-positive species were determined by enzyme linked immunosorbent assay (ELISA) using 96-well ELISA Microplates (BRAND immunoGrade, 781722). Plates were coated overnight at 4 °C with 50 µl of the respective samples diluted 1:100 in coating buffer [15 mM Na$_2$CO$_3$ (Roth, 8563.1), 35 mM NaHCO$_3$, pH 9.6 (Merck, 1.06329.1000). Commercially available purified LTA from Bacillus subtilis (Sigma-Aldrich) served as standard and was coated in duplicate with 2-fold dilutions starting at 1 µg/ml in coating buffer. After washing three times with 0.05% (v/v) Tween-20 in PBS (PBS-T), nonspecific binding sites were blocked with 1% bovine serum albumin (BSA, Roth, 8076.4) in PBS (PBS-B, 200 µl per well) for 2 h. Plates were washed three times with PBS-T, incubated for 1 h with 100 µl mouse anti-LTA G43J (1:500 in PBS-B, Thermo Fisher Scientific, MA1-7402) per well, again washed three times with PBS-T and incubated for 1 h with 100 µl of HRP-conjugated goat anti-mouse (1:10,000 dilution in PBS-B, Jackson ImmunoResearch Laboratories, JIM-115-035-003) per well. After five washes with PBS-T, plates were developed using the TMB substrate (Biolegend, 421101) and stop solution was added [1 M H$_2$SO$_4$ (Roth, 9316.1)] according to the manufacturer's instructions. Optical densities were read at 450 nm with a SPECTROstar nano plate reader (BMG Labtech, Germany).

## Lipid quantification

Lipid quantification was performed using FM 4-64 (N-(3-Triethylammoniumpropyl)-4-(6-(4-(Diethylamino) Phenyl) Hexatrienyl) Pyridinium Dibromide, Thermo Fisher Scientific, T13320) fluorescent dye as according to previous reports as follows[72,73]:. FM 4-64 was dissolved in PBS to a working solution of 0.2 µg µl$^{-1}$. 4 µl of FM 4-64 solution was mixed in a 96-well microtiterplate (Greiner Bio-One, PS, F-bottom, Black, Cellstar, 655086) with 200 µl of PEF derived from human colonic luminal samples, BEVs and hEVs (1 µg protein equivalent). PEF, BEVs and hEVs without FM 4-64 dye as well as FM 4-64 dye diluted in PBS served as controls. Following a 10 min incubation at 37 °C, endpoint fluorescence (excitation: 485 nm/ emission: 620 nm) was analyzed using the FLUOStar Omega microplate reader (BMG LabTech). Data is presented as ratio calculated as fluorescence (RFU) per µg protein in BEV sample.

## Nucleic acid quantification

PEFs derived from human colonic luminal samples, BEVs and hEVs associated nucleic acid was quantified using membrane-permeable nucleic acid dye SYTO 9. Chromosomal DNA isolated from *V. cholerae* was used as standard. 45 µL of an appropriate dilution of PEF or BEV sample in saline were mixed with 5 µl of SYTO 9 green fluorescent nucleic acid stain (5 mM, Thermo Fisher Scientific, S34854) in a 96-well microtiter plate (Greiner Bio-One, PS, F-bottom, Black, Cellstar, 655086). The mixture was incubated in dark for 1 h at room temperature. Afterwards, the endpoint fluorescence (excitation: 485 nm/ emission: 520 nm) was measured using the FLUOStar Omega microplate reader (BMG LabTech). Measured fluorescence was used to back-calculate the nucleic acid concentration of each sample using the standard curve obtained from the chromosomal DNA standard, with the results expressed in nucleic acid amount in ng per µg protein.

## Dot-blot

Dot blot analysis to compare marker signal intensities from a known amount of hEV derived from HT-29 to that in PEFs was essentially performed according to previous reports as follows[74,75]: CD9, CD81 and CD63 were used as bona fide hEV biomarkers according to MISEV 2023[19]. Briefly, four 2-fold dilutions starting at 2 µg (protein equivalent) of PEF or 1 µg (protein equivalent) of HT-29 derived hEVs (serving as standard) were spotted onto nitrocellulose membranes using a 96-well Whatman Minifold I Dot Blot System. In addition, each membrane was spotted with 1% BSA (Roth, 8076.4) as a negative control and with commercially available purified CD9 (40 ng) (Sino Biological, 11029-H08H) or CD81 (40 ng) (Sino Biological, 14244-HNCH) as positive control if applicable. Then, each membrane was dried, incubated in Tris-buffered saline (TBS) [20 mM Tris/HCl (Roth, 4855.2), pH 7.5, 150 mM NaCl (Roth, 3957.1)] for 2 min, and blocked in 10% skim milk (Roth, T145.2) in TBS (TBS-S) for 2 h. Afterwards two membranes spotted with the same samples were either incubated for 2 h with mouse anti-CD9 (Thermo Fisher Scientific, 10626D), rabbit anti-CD81 (Thermo Fisher Scientific, 10630D) or mouse anti-CD63 (Thermo Fisher Scientific, 10628D) antibody 1:500 diluted in TBS-S. Membranes were washed three times in TBS for 10 min, incubated with the respective secondary antibody [HRP-conjugated anti-mouse IgG (Jackson ImmunoResearch Laboratories, JIM-115-035-003) or HRP-conjugated anti-rabbit IgG (Jackson ImmunoResearch Laboratories, JIM-111-035-003), each diluted 1:2,000 in TBS-S] for 1 h, washed once in TBS-T and twice in TBS for 10 min each. Chemiluminescence was detected via the Immun-Star WesternC Kit (Bio-Rad Laboratories, 170-5070) in combination with ChemiDoc XRS system (Bio-Rad Laboratories) and the Quantity One software (Bio-Rad Laboratories). Quantification was performed on acquired images with exposure times just before reaching saturation (see Supplementary Fig. 3h for representative examples of dot blots). Relative intensities of unsaturated spots were evaluated by the Quantity One software (Bio-Rad Laboratories) using the volume circle tool. The relative intensities of spots derived from the four dilutions of HT-29 hEVs were used to generate a polynomial regression curve, which was subsequently used to calculate the biomarker amount of hEVs (CD9, CD81 or CD63) present in the PEF samples given by the relative intensity of the most concentrated, non-saturated sample spot. Data for each PEF sample is presented as percentage as follows: (hEV biomarker amount of determined by dot blot)/ (total protein amount determined by Bradford assay) *100.

## IgA-coating of BEVs and hEVs

150 µg protein equivalent of BEVs or hEVs were diluted in saline to final volume of 300 µl and mixed with 100 µl of an IgA blend containing 50 µg of IgA from human serum (Sigma-Aldrich, I4036) and 50 µg of IgA from human colostrum (Sigma-Aldrich, I2636). In parallel, 150 µg protein equivalent of BEVs or hEVs were diluted in saline to final volume of 300 µl and mixed with 100 µl saline without IgA to generate similar treated uncoated EV samples. After incubation on a laboratory rotator at 4 °C for 24 h sample volume was increased to 5 ml with saline and subjected to ultracentrifugation (90,000 g, 4 °C, 4 h,). After one

time washing with 5 ml saline the pellet containing IgA-coated or uncoated BEVs or hEVs was resuspended in 100 µl saline and stored at −20 °C until further use.

## Immunoglobulin quantification

IgA and IgG levels in soluble and PEFs derived from human colonic luminal samples as well as BEVs and hEVs after IgA-coating were determined by ELISA essentially as described above ("LTA quantification") with the following modifications: Commercially available purified human IgA (Sigma-Aldrich, I4036) or IgG (Sigma-Aldrich, I2511) served as standards and were coated in duplicate with 2-fold dilutions starting at 100 ng ml⁻¹ in coating buffer. HRP-conjugated goat anti-human IgA (1:1,000 diluted in PBS-B, Biolegend, 411002) or HRP-conjugated donkey anti-human IgG (1:5000 diluted in PBS-B, Biolegend, B410902) served as detection antibodies thereby avoiding the use of a second antibody.

## Detection of sCD89-IgA, IgA-LTA and IgA-OmpA complexes

Amount of IgA bound to CD89, LTA and OmpA was detected by sandwich ELISA using 96-well ELISA Microplates (BRAND immuno-Grade, 781722). Plates were coated overnight at 4 °C with 100 µl of mouse anti-human CD89 (1 µg ml⁻¹, Bio-Rad Laboratories, MCA1824), mouse anti-LTA (0.5 µg ml⁻¹, Thermo Fisher Scientific, MA1-7402) or rabbit anti-OmpA (0.9 µg ml⁻¹, Abbexa, abx110631) antibodies diluted to the indicated final concentrations in coating buffer. Commercially available purified human serum IgA (Sigma-Aldrich, I4036) served as standard and was coated in duplicate with 2-fold dilutions starting at 100 ng ml⁻¹ in coating buffer. After washing three times with PBS-T, nonspecific binding sites were blocked with PBS-B (200 µl per well) for 2 h, before plates were washed three times with PBS-T and respective wells were incubated for 2 h with 100 µl soluble and PEFs derived from human colonic luminal samples 1:100 diluted in PBS. After washing three times with PBS-T, wells were incubated with 100 µl of HRP-conjugated goat anti-human IgA (0.4 µg ml⁻¹ in PBS-B, Biolegend, 411002) for 1 h. After five washes with PBS-T, plates were developed using the TMB substrate (Biolegend, 421101) and stop solution [1 M $H_3SO_4$ (Roth, 9316.1)] according to the manufacturer's instructions. Optical densities were read at 450 nm with a SPECTROstar nano plate reader (BMG Labtech, Germany).

## Enrichment of IgA-coated EVs in human pelletable effector fractions by immunoprecipitation

IgA-coated EVs in PEF from three randomly chosen active UC patients were enriched using the Dynabeads Protein G immunoprecipitation kit (Thermo Fisher Scientific, 10007D) according to the manufacturer's protocol with minor modifications. Dynabeads magnetic beads were coated with 5 µg of rabbit anti-human IgA antibody (Sigma-Aldrich, SAB5600221) for 10 min at room temperature, washed with washing buffer and finally recovered by pulldown using a magnetic separation rack (NEB, S1506S). Appropriate volumes of the PEF samples (100 µg) were then added to anti-human IgA-coated magnetic beads and incubated for 20 min at room temperature on a lab rotator. Afterwards, the magnetic beads were washed three times with washing buffer to remove unbound material and finally recovered by pulldown using a magnetic separation rack (NEB). Transfer into new reaction tubes excluded contamination of non-specifically bound material to the plastic wall of the reaction tubes. Immunoprecipitated effector complexes were eluted from the anti-human IgA-coated magnetic beads by elution buffer and subjected to NTA and mass spectrometry analysis or stored at −20 °C until further use.

## Quantitative proteomics via mass spectrometry (MS)

Samples before and after immunoprecipitation were analyzed by quantitative proteomics via mass spectrometry (MS) at the Functional Genomic Center Zurich (FGCZ), University of Zurich. Label-free quantitative proteomics was performed employing the filter-aided sample preparation (FASP) method according to previous reports as follows[76]: 20 µg protein equivalent [quantified using the Qubit Protein Assay kit (Thermo Fisher Scientific)] for each human PEF before and after immunoprecipation (for details see chapter above) was denatured in 4% SDS, 0.1 M DTT, 8 M urea, 100 mM Tris, pH 8.2 at 95 °C for 5 min and subjected to IAA alkylation (50 mM) on a Microcon filter unit (Millipore). Samples were then washed with urea buffer (8 M urea, 100 mM Tris, pH 8.2) and 0.5 M NaCl, and further digested with 0.4 µg trypsin (Promega) overnight at room temperature. Digested peptides were then eluted (50 mM Triethylammonium bicarbonate buffer, pH 8.5), acidified (final concentration of 0.5% trifluoracetic acid) and desalted with C18 stage tips[77]. The samples were freeze dried and stored in −20 °C prior to LC-MS analysis. For LC-MS data acquisition, iRT peptides (Biognosys) were added to the samples for calibration. Peptides were separated on an ACQUITY UPLC M-Class System (Waters) equipped with a HSS T3 C18 reverse-phase column (1.8 µm, 75 µm × 250 mm, Waters) and analyzed by an Orbitrap Fusion Lumos Tribrid mass spectrometer (Thermo Fisher Scientific). Data were acquired in DDA mode with solvent A (0.1% formic acid in H2O) and solvent B (0.1% formic acid in acetonitrile) using a 108 min gradient; 5% B for 3 min, 5–22% in 80 min, 22–32% in 10 min, 32–95% in 5 min, 95% for 10 min with column temperature 50 °C and a constant flow rate of 0.3 µl min⁻¹. Thermo raw files were converted to the Mascot generic format by the Proteome Discoverer, v2.0 (Thermo Fisher Scientific) using the automated rule-based converter control[78]. Mascot searching was performed against a database including *Ruminococcus gnavus* CAG126, *Bacteroides fragilis* CL07T12C05, *Bifidobacterium bifidum* BGN4, *Enterobacter cloacae* subsp cloacae, *Enterococcus faecalis* (strain ATCC 700802 V583), *Escherichia coli* O157H7, *Faecalibacterium prausnitzii*, *Klebsiella pneumoniae* IS43, *Lactobacillus acidophilus* (strain ATCC 700396 NCK56 N2 NCFM), *Lactococcus lactis* subsp. lactis, *Proteus mirabilis* (strain HI4320), *Pseudomonas aeruginosa* (strain ATCC 15692 DSM 22644 CIP 104116 JCM 14847 LMG 12228 1 C PRS 101 PAO1), *Saccharomyces cerevisiae* (strain ATCC 204508 S288c), *Salmonella enterica* subsp enterica serovar Poona, *Staphylococcus aureus* (strain NCTC 8325 PS 47), *Streptococcus thermophilus* CNCM I-1630, and *Homo sapiens* obtained from Uniprot combined with common contaminants and decoys with parameters as follows; fixed modification carbamidomethyl (C) and variable modification deamidated (NQ) and oxidation (M); max cleavage 1; peptide charges 2 +, 3 +, 4 +; peptide tolerance 10 ppm; MS/MS tolerance 0.6 Da. Progenesis QI software v4.2 (Waters) was used for quantitative proteomic analysis. Alignment of chromatograms was carried out by combining automatic and manual alignment with iRT peptide standards. Peptides with MS/MS spectra ranking greater than 6 were excluded. Peptide and protein identification was done by Scaffold 5 (Proteome Software) with Mascot searching and was imported back to Progenesis. BamA, BamC, Lpp, TolB, TolC, and their homologs were selected as bacterial EV biomarkers across all the species in the database used for Mascot search. Normalized spectral counts of each selected protein was calculated into percentile individually and applied with conditional formatting with a 3-color scale by Excel.

## Nanoparticle tracking analysis (NTA)

The concentration and size distribution of particles in the PEFs of colonic luminal samples, BEVs and hEVs were evaluated using a NanoSight NS300 instrument [Malvern Panalytical Ltd, UK located at the University of Zurich (comparative scatter and fluorescent mode measurement) or Medical University of Graz (scatter mode measurement for concentration and size distribution)] with NTA 3.4 software. For fluorescent labeling of EVs, 10 µg protein equivalent of the respective samples were diluted in PBS to a final volume of 100 µl, mixed with 2 µl mouse anti-LTA G43J antibody (Thermo Fisher Scientific, MA1-7402) or 1 µl rabbit anti-human IgA antibody (Sigma-Aldrich,

SAB5600221) and incubated for 6 h at 4 °C with gentle agitation (300 rpm). Afterwards 0.5 µl of Alexa Fluor Plus 488 secondary antibody [donkey anti-mouse IgG (H + L), Jackson ImmunoResearch Laboratories, 715-545-150, or goat anti-rabbit IgG (H + L), Thermo Fisher Scientific, A32731] was added and incubated ON at 4 °C with gentle agitation (300 rpm). Samples were then diluted in HyClone HyPure Water (Cytiva, SH30538.03) to a final concentration of 20–50 particles per frame before being measured by scattering or fluorescence (filter 500 nm) mode with a laser 488 module. For each sample, ten 60 s videos were recorded in the camera level 16 and analyzed with threshold 5. Pulldown efficiency of EV fractions derived from active UC patients (three randomly chosen representatives) before and after immunoprecipitation by anti-human IgA antibody was calculated as follows: particle amount detected in the fluorescent mode / total particle amount detected in scattering mode * 100 %.

## Transmission electron microscopy

To visualize EVs in PEF of colonic luminal samples, BEVs and hEVs by transmission electron microscopy (TEM) appropriate dilutions in PBS (Sigma-Aldrich) were allowed to adsorb on a pre-glow-discharged (15 mA, 25 s, PELCO easiGlow) formvar-coated 300-mesh copper grid (Plano GmbH, SF162-3). After 1 min, the excess liquid was removed using filter paper. The grid was negatively stained with 1% uranyl acetate (Honeywell Fluka, 73943) in case of BEVs and EVs in PEF of colonic luminal samples or 2% uranyl acetate in case of hEVs for 1 min and dried with filter paper.

In case immunogold co-staining, PEF of colonic luminal samples or BEVs were adjusted to 1 µg µl⁻¹ (protein equivalent) in PBS (Sigma-Aldrich) with 1% glutaraldehyde (Sigma-Aldrich, G5882), allowed to absorb on pre-glow-discharged (15 mA, 25 s, PELCO easiGlow) formvar-coated 300-mesh copper grid (Plano GmbH, SF162-3) for 5 min, thoroughly rinsed with PBS, incubated with 0.05 M glycine (Roth, 3790.2) for 10 min and subsequently with PBS containing 3% skim milk (Roth, T145.2) for 30 min to reduce unspecific binding. Afterwards the grids were incubated overnight at 4 °C in PBS with 3% skim milk containing goat anti-human IgA-antibody conjugated with 20 nm gold particles (Cytodiagnostics, A-20-27-15; diluted in a range of 1:200 to 1:500), rabbit anti-OmpA antibody (Abbexa, abx110631, diluted 1:100) and a mouse anti-LTA antibody (Thermo Fisher Scientific, MA1-7402, diluted 1:200). After rinsing 5 times with PBS containing 3% skim milk for 5 min the grids were incubated for 1 h at room temperature in PBS with 3% skim milk containing the secondary antibodies, i.e., preadsorbed donkey anti-rabbit IgG conjugated with 6 nm gold particles (abcam, ab105294, diluted 1:20) and preadsorbed donkey anti-mouse IgG with 6 nm gold particles (abcam, ab105276, diluted 1:40). The grids were then rinsed again 5 times with PBS containing 3% skim milk for 5 min and finally rinsed 2 times in sterile $H_2O$ (Fresenius, 20127078200) for 10 min. After drying for 30 min the grids were negatively stained with 1% uranyl acetate (Honeywell Fluka, 73943) for 1 min. The ideal dilutions of all antibodies were determined in preliminary experiments by evaluating the labeling density after a series of labeling experiments. The final dilution of primary and secondary antibodies used in this study showed a minimum of background labeling outside the EVs with a maximum of specific labeling attached to the EVs. Uncoated BEVs served as negative control to ensure the specific binding of the 20 nm gold particle-conjugated anti-human IgA-antibody.

Samples were visualized using a Zeiss Libra 120 Plus TEM (Carl Zeiss AG, Oberkochen, Germany) and micrographs were recorded with a XF416 4k camera (Tietz Video and Image Processing Systems GmbH, Gauting, Germany).

## Effector stimulation of HT-29 and U937 cells

For cytokine assay with HT-29 cells, $6 \times 10^5$ cells ml⁻¹ were seeded in 24-well tissue culture plates and cultivated for 24 h in DMEM/F-12 medium (Thermo Fisher Scientific, 21331-020) with FBS (Thermo Fisher

Scientific, 10270-106). Cells were washed once with PBS and the medium was replaced with SEF or EV fractions from human endoscopic samples (-0.007 µg ml⁻¹ protein equivalent for SEF and 0.002 µg ml⁻¹ for EV fractions) or defined amounts of BEVs (as indicated) resuspended in DMEM/F-12 without FBS. Mock-treated cells cultivated in growth medium served as controls. After incubation for 16 h, supernatants were removed and analyzed by ELISA or stored at −20 °C until further use.

In case of U937, $2.5 \times 10^5$ cells ml⁻¹ were differentiated to CD89⁺CD14⁺CD11b⁺ monocytes by addition of vitamin D3 (100 nM, 1α,25-Dihydroxyvitamin D3, Sigma-Aldrich, D1530) and TGF-β1 (0.5 ng ml⁻¹, rhTGF-β1, R&D Systems, 240-B-010) for 48 h For assessment of cytokine production U937 monocytes or undifferentiated U937 cells were resuspended in RPMI 1640 (Sigma-Aldrich, R8758) without FBS and seeded in 24-well tissue culture plates at a concentration of $2 \times 10^5$ cells ml⁻¹ in presence of soluble or pelletable fractions from human endoscopic samples (-0.007 µg ml⁻¹ protein equivalent for SEF and 0.002 µg ml⁻¹ for EV fractions in case of IL-8 and 0.07 µg ml⁻¹ protein equivalent for SEF and 0.02 µg ml⁻¹ for EV fractions in case of IL-6), defined amounts of BEVs (as indicated) or 1 µg of peptidoglycan (Sigma-Aldrich, 69554/ 77140). Mock-treated cells cultivated in growth medium served as controls. After incubation for 16 h the culture supernatants were removed, subjected to centrifugation (10 min, 4 °C, 1125 $g$) to remove cellular debris and analyzed by ELISA or stored at −20 °C until further use.

## Human organoid-derived monolayers and effector stimulation

Colonic organoids were established according to previous reports as follows[79,80]: Crypt structures were squeezed out of meshed up biopsy pieces, embedded into basement membrane extract (BME, BioTechne) and cultured in colon medium[79] (Advanced DMEM/F12 supplemented with glutamine, HEPES and Pen/Strep, 20% R-spondin conditioned medium [self-made], 1% Noggin [ImmunoPrecise], 0.5 nM WNT surrogate [ThermoFisher], B27, 10 mM nicoti-naminde, 1.25 mM N-Acetyl Cysteine, 50 µg/ml EGF [Peprotech], 500 nM A83-01, 10 nM Prostaglandine E2, 1 µM P38 inhibitor and Primocin). Cultures were passage once per week by mechanical shearing and RhoK inhibitor (Y-27632) was added directly after seeding. Medium was exchanged twice per week. For the establishment of organoid-derived monolayers on transwells, a single cell suspension of organoid culture was generated by using TrypLE (Thermo Fisher) and 250,000 cells were seeded on BME-coated 24-well transwells. Confluent cultures were obtained after three to four days. Monolayers were then exposed to randomly selected samples of EV fractions from human endoscopic samples (-0.1 µg ml⁻¹ protein equivalent) or BEVs (as indicated) the apical side. After 24 h, the liquid of the basal compartment was collected, potential cell debris removed by a 400 $g$ centrifugation before organoid-derived monolayer-secreted IL-8 levels were measured by ELISA. The integrity of monolayer cultures was recorded by microscopy and the cell viability measured in a CellTiterGlo (Promega) assay according to manufacturer's protocol.

## Isolation and effector stimulation of primary human monocytes isolated from peripheral blood

CD14⁺ monocytes were isolated from buffy coat preparations obtained from the Department for Blood Serology and Transfusion Medicine, Medical Hospital Graz. CD14⁺ cells were positively selected from MNC fractions (prepared with Lymphoprep (Axis Shield, #1114547) with magnetic beads using the Human CD14 MicroBeads isolation kit (Miltenyi, #130-050-201) according to the manufacturer's protocol. IgA-coated and uncoated BEVs as well as randomly selected samples of EV fractions from human endoscopic samples were tested on CD14⁺ monocytes. CD14⁺ monocytes were seeded in 24-well tissue culture plates at concentration of $2 \times 10^5$ cells ml⁻¹ in presence of EVs fractions (-0.015 µg ml⁻¹ protein equivalent) or defined amounts of BEVs (as

indicated). After 16 h of incubation, subjected to centrifugation (10 min, 4 °C, 1125 $g$) to remove cellular debris and analyzed by ELISA or stored at −20 °C until further use.

## Quantification of human cytokine responses

IL-6 and IL-8 levels in the cell culture supernatants were quantified by ELISA using 96-well ELISA Microplates (BRAND immunoGrade, 781722) according to the manufacturer's recommendations. Plates were coated overnight at 4 °C with the designated capture antibody (2.4 μg ml⁻¹, 100 μl per well, Biolegend, 501102 & 514602) diluted in coating buffer. After washing three times with PBS-T, nonspecific binding sites were blocked with PBS-B (200 μl per well) for 2 h, before plates were washed three times with PBS-T and wells were incubated either with respective cytokine standards or cell culture supernatant samples (100 μl per well) for 2 h. Commercially available recombinant human IL-6 and IL-8 (Biolegend, 570809 & 570909) served as standard and were used in duplicates with 2-fold dilutions starting at 1000 pg ml⁻¹ in coating buffer. After washing three times with PBS-T, plates were incubated with the respective biotin-conjugated detection antibody (2.4 μg ml⁻¹, 100 μl per well, Biolegend, 501202 & 514704) diluted in PBS-B for 1 h, washed three times with PBS-T and incubated with the avidin-HRP-conjugate (1:1000, 100 μl per well, Biolegend, 405103) diluted in PBS-B for 30 min. After five washes with PBS-T, plates were developed using the TMB substrate (Biolegend, 421101) and stop solution [1 M $H_3SO_4$ (Roth, 9316.1)] according to the manufacturer's instructions. Optical densities were read at 450 nm with a SPECTROstar nano plate reader (BMG Labtech, Germany).

## Transgenic mouse model expressing human CD89

Myeloid-specific CD89-expressing mice were generated by crossing CD89tg/wt (obtained from CSL Limited, Australia) and C57BL/6 LyzMcre/cre Tg mice (licensed from Jackson Laboratories, MGI:1934631) to excise the loxP-flanked mCherry cassette in vivo as previously reported[51]. Excision of the mCherry cassette results CD89 expression in the myeloid cell lineage under a CMV promoter in 50% of the offspring (CD89tg/wt/LyzMcre/wt; CD89-transgenic mice). Littermates (CD89wt/wt/LyzMcre/wt; LM) not expressing CD89 served as CD89-negative controls. The CD89 gene-containing offspring were detected by PCR using the transgene-specific primers CD89tg P1 and CD89tg P2 (Supplementary Table 4).

## Murine colitis model

7–8 week old female mice (CD89-transgenic mice and littermates) were given 2% Dextran Sodium Sulfate salt (DSS, MP Biomedicals, 216011090) in drinking water ad libitum for 4 days to induce colitis, on the 5th day DSS was removed and replaced by regular water ad libitum.

On day 4 to 7 mice were orally gavaged twice a day (morning and evening) with 100 μl of IgA-coated bacterial BEVs^B/L (0.5 μg μl⁻¹ protein equivalent), uncoated bacterial BEVs^B/L (0.5 μg μl⁻¹ protein equivalent) or saline. On day 8 mice were only gavaged once in the morning. On day 9 mice were euthanized by cervical dislocation, cecal luminal contents were collected for lipocalin extraction, colons were removed, analyzed for length, embedded as swiss-rolls and fixed in 4% formalin for blinded histo-pathological evaluation.

## Isolation of mouse bone marrow derived cells (BMDCs), effector stimulation, Syk inhibition and quantification of cytokine responses

BMDCs were isolated from mouse bone-marrow precursors. In brief, femur and tibiae were removed from 7 to 8-week-old mice (CD89-transgenic mice and littermates). Bone marrow cells were flushed out with RPMI 1640 medium (Sigma-Aldrich, R8758) supplemented with 1% penicillin/ streptomycin (Thermo Fisher Scientific, 15140-122) and 10% FBS (Thermo Fisher Scientific, 10270-106), purified through a 100 μm cell strainer (Corning, CLS431752-50EA) and incubated at 37 °C and 5%

$CO_2$ in a humidified atmosphere for 24 h. On the second day, $2 \times 10^5$ BMDCs were seeded in 24-well tissue culture plates in presence of IgA-coated and uncoated BEVs resuspended in RPMI 1640 medium or in presence of randomly selected samples of EV fractions from human endoscopic samples (~0.004 μg ml⁻¹ protein equivalent). Mock-treated cells cultivated in growth medium served as controls. After incubation for 16 h the culture supernatants were removed, subjected to centrifugation (10 min, 4 °C, 1125 $g$) to remove cellular debris and analyzed by ELISA or stored at −20 °C until further use. For inhibition assays of functional Syk, which represents an essential downstream factor in the CD89-dependent signaling pathway, $2 \times 10^5$ BMDCs were isolated from CD89 mice as described above, preincubated for 1 h with 1 μM R406 (Syk inhibitor, InvivoGen, inh-r406n) and then stimulated with EV fractions from active UC patients (approx. 0.004 μg ml⁻¹ protein equivalent) and IgA coated BEVs ^B/L (5 μg ml⁻¹). After 16 h of incubation, subjected to centrifugation (10 min, 4 °C, 1125 $g$) to remove cellular debris and analyzed by ELISA or stored at −20 °C until further use.

IL-6 and TNF-α mouse uncoated ELISA Kits (Thermo Fisher Scientific, 88-7064-88 & 88-7324-77) in combination with ELISA Microplates (BRAND immunoGrade, 781722) were used to quantify the pro-inflammatory cytokine responses according to the manufacturer's protocol.

## Histopathological analysis

Colon swiss roles were embedded in paraffin and sections were stained with hematoxylin and eosin according to standard methods. Sections were assessed and scored with a light microscope (Nikon Eclipse Ni) and measurements (swiss-role total length, ulcer/erosion length) were performed with the adjoined microscopy camera and the imaging software NIS Elements (version 5.2; Nikon). A semi quantitative 3-tier scoring system was used for colitis scoring (0 = absent, 1 = focal, 2 = present, 3 = abundant) encompassing epithelial alteration, abundance of neutrophils, presence of ulcers/erosions yielding a total histology score ranging from 0 to 9.

## Murine lipocalin and human IgA quantification in mouse cecal content

Mouse lipocalin was analyzed from the mouse cecal content collected from each mouse directly after euthanization by cervical dislocation. Soluble material including lipocalin was extracted from cecal content by adding 800 μl PBS per 100 mg cecal weight. After vortexing the samples for 15 min at 4 °C, solid material was separated by centrifugation (10 min, 13,000 $g$) and the supernatants were stored at −80 °C. Lipocalin-2 levels were quantified by ELISA according to the manufacturer's protocol (Mouse Lipocalin-2/NGAL DuoSet ELISA, R&D system, DY1857-05). Human IgA was quantified in the mouse cecal content collected from mice receiving human IgA-coated and uncoated BEVs and processed as described above. Human IgA was detected by ELISA using 96-well ELISA Microplates (BRAND immunoGrade, 781722). Plates were coated overnight at 4 °C with 50 μl of cecal content sample. For details on the ELISA see section "Immunoglobulin quantification".

## Flow cytometry

U937 cells and mouse BMDC were collected, washed and resuspended in PBS. Staining was performed at 4 °C using the antibodies anti-human CD11b-PE-Cy7 (Biolegend, 301322, diluted 1:200), anti-human CD14-AF700 (BD Bioscience, 557923, diluted 1:50), and anti-human CD89-PE (BD Bioscience, 555686, diluted 1:50). 7-AAD (BD Bioscience, 559925, 5 μl/sample) was used as viability stain and added 5 min prior to analysis. Flow cytometry experiments were measured with a LRS Fortessa flow cytometer with FACSDiva 9.0.1 (BD Biosciences). Data were analyzed using FlowJo 10 (FlowJo, LLC) software at the Medical University of Graz. In case of organoids, a single cell suspension of TrypLE dissociated organoid cultures was stained for 20 min at 4 °C with anti-human CD89-PE (BD Bioscience, 555686, 1:50). Data were then

acquired on a CytoFlex machine and subsequently analyzed using FlowJo software at the CeMM, Vienna. Respective gating strategies are exemplified in Supplementary Fig. 3a and Supplementary Fig. 7.

## Immunofluorescence

Studies were carried out on 4% paraformaldehyde (PFA)-fixed, paraffin embedded human colonic biopsies and mouse intestinal tissue samples, sectioned at 5 μm. Slides were deparaffinized in xylene, and rehydrated with decreasing concentrations of ethanol according to standard methods. Tissue sections were submerged to HIER antigen retrieval in Target Retrieval Solution pH 6.0 (Agilent/Dako, S3022) for 20 min in a domestic microwave oven. Slides were allowed to cool for 45 min at room temperature before rinsing in wash buffer Tris-Buffered Saline Tween-20 (TBST, pH 7.4). Sections were blocked with 5% donkey serum (Jackson ImmunoResearch Laboratories, 017-000-001) + 5% BSA (Sigma-Aldrich, A9418) in TBST for 1 h prior to primary antibodies incubation at 4 °C overnight. Negative controls were incubated with the appropriate IgG fractions as isotope controls. All incubation steps were performed in a dark moist chamber at room temperature. After 10 min of TBST wash secondary antibodies were applied for 30 min. Followed by another TBST wash, DAPI (Thermo Fisher Scientific, D1306, 0.75 μg ml$^{-1}$) was added to the slides for 15 min as a nuclei counter stain. Sections were rinsed again with TBST before mounting with Vectashield mounting medium (Vector Lab, Inc., USA, VECH-1000).

The following primary antibodies were used: monoclonal rabbit anti-human CD89 (Abcam USA, ab124717, 0.5 μg μl$^{-1}$), monoclonal rat anti-mouse CD11b/ITGAM (M1/70) (Cell Signaling USA, 46512, 1.25 μg ml$^{-1}$), monoclonal mouse anti-human CD68 (clone Kp1) (Dako-Cytomation, M0814, 1.17 μg ml$^{-1}$), negative rabbit IgG (Calbiochem, NI01), mouse IgG 1 control (Calbiochem, NI03), and rat IgG2b (Invitrogen, 14-4321-81). Cy-3 donkey anti-rabbit IgG (H + L) (Jackson ImmunoResearch Laboratories, 711-165-152, 3.5 μg ml$^{-1}$) and Alexa Fluor 488 donkey anti-rat IgG (H + L) (Jackson ImmunoResearch Laboratories, 712-545-150, 1.4 μg ml$^{-1}$) and Alexa 488 donkey anti-mouse IgG (H + L), (Jackson ImmunoResearch Laboratories, 715-454-150, 1.75 μg ml$^{-1}$) were used as secondary antibodies. Nuclei were counterstained with DAPI (Thermo Fisher Scientific, D1306, 0.75 μg ml$^{-1}$). All antibodies were diluted with Dako Cytomation Antibody Diluent with Background Reducing Components (Dako, S3022).

To acquire and analyze computerized images of sections, a Leica DM4000 B microscope (Leica Cambridge Ltd) equipped with Leica K5 Video camera (Leica Cambridge Ltd) were used.

## Statistical analysis

Patients were recruited prospectively for the individual groups. As no a priori comparison and hypothesis were made on group differences, no statistical methods were used to pre-determine sample sizes. However, our sample sizes are similar to those reported in previous publications[29–32]. The sample size was sufficient to demonstrate statistical significance in comparisons between the different experimental groups (non-IBD versus inactive/ active UC). With exception of their genotype, mice were randomly assigned to respective treatment groups and histological scoring was done double-blinded. Data collection, statistical analysis and visualization utilized Microsoft Excel (16.16.27) GraphPad Prism (8.4.0) and CorelDRAW 2021 (23.1.0.389). We generally assumed non-normal distribution, but this was not formally tested. Testing for statistical significance always employed two-tailed tests if not stated otherwise. Details of the statistical tests used are provided in the manuscript and figure legends. $P$ values less than or equal to 0.05 were considered significant.

## Reporting summary

Further information on research design is available in the Nature Portfolio Reporting Summary linked to this article.

## Data availability

Whenever possible the data generated in this study are provided in the Source Data file. The clinical data is provided as pseudonymized data in the Source data file. The FACS and image data generated in this study have been deposited in the public repository https://zenodo.org under accession codes https://doi.org/10.5281/zenodo.15019428 [https://zenodo.org/records/15019428] and https://doi.org/10.5281/zenodo.15023687 [https://zenodo.org/records/15019428]. Details on the clinical data and additional human cohort information are available under restricted access for reasons of sensitivity, access can be obtained from the corresponding author (christoph.hoegenauer@medunigraz.at) upon reasonable request, which will be responded within 8 weeks. Source data are provided with this paper.

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

## Acknowledgements
We are grateful to CSL Limited and its representative I. Campbell for providing the CD89 mouse line and assistance along the transfer. We thank K. Zierler and the entire in-house animal facility (Institute of Molecular Biosciences, University of Graz) for maintenance of the mouse group, H. Strohmaier and the Imaging Core facility (Medical University of Graz) for access to the NS300 as well as A. Streit and A. Posch as well as the endoscopy unit team (Division of Gastroenterology and Hepatology, Medical University of Graz) for their help in obtaining the clinical samples. TEM imaging was carried out at the section for Plant Cell Biology at the Institute of Biology, University of Graz. We thank Stefan Möstl and Prof. Johannes Liesche for technical assistance. Figures 1a, 7a and Supplementary Fig. 6b have been created with BioRender.com. This research was funded in whole, or in part, by the Austrian Science Fund (FWF) [grants P33073 to S.S; DOC50 docfund "Molecular Metabolism" to S.S.], by the BioTechMed-Graz Flagship Project Secretome to S.S. & H.S., the DocAcademy Graz to S.S. as well as the Land Steiermark and the City of Graz. For the purpose of open access, the author has applied a CC BY public copyright licence to any Author Accepted Manuscript version arising from this submission. The authors acknowledge the financial support by the University of Graz.

## Author contributions
The manuscript was initially drafted by H.B.T. and S.S. and further developed by C.H. and E.L.Z. H.B.T., C.H. and S.S. designed the concept of the work. All authors contributed to the conception and design of the study along the individual experiments. The clinicians A.B, L.B., and C.H. were responsible for patient enrollment, collection of colonic luminal samples and assessment of disease activity parameters. H.B.T. and P.K. established and performed human sample fractionation and H.B.T. performed bacterial MV isolation, which included subsequent sample characterization and pro-inflammatory response profiling. J.S. assisted in the immunoglobulin titer quantification of samples. H.B.T. and C.A.P. established the U937 cell line protocols and performed FACS under supervision of H.S., C.T.-A. and C.A.P. established and performed immunofluorescence analyses under supervision of H.S. Nanosight analyses and mass spectrometry Y.-C.C., R.K. and D.F. under supervision of L.E. Electron microscopy was initially established by R.K. and D.F. and finally performed by K.S.-P. and D.F. Organoid experiments including their FACS analysis were performed by M.R.G. under supervision of G.A.B. G.G. performed histological analyses of mouse tissue samples. Mouse experiments were performed by H.B.T., D.F., and S.S. All authors contributed to critical review and revision of the manuscript. All authors had full access to all the data in the study and had final responsibility for the decision to submit for publication.

## Competing interests
The authors declare no competing interests.
