## [Transparent Peer Review file · Nature Communications]

Enrichment of human IgA-coated bacterial vesicles in ulcerative colitis as a driver of inflammation

Corresponding Author: Professor Stefan Schild

Version 0:

Reviewer comments:

Reviewer #1

(Remarks to the Author)

In this manuscript, Thapa et al. report on their studies on IgA-coated bacterial microvesicles in the context of and as a driver of ulcerative colitis. The authors suggest that IgA-coated bacterial microvesicles are increased in active UC and have an important pro-inflammatory function by activating CD89+ immune cells to produce cytokines.

Although this is conceptually appealing and might help to explain previously unrecognized aspects of microbiota-driven IBD pathogenesis, several aspects limit the significance of the findings.

Major points:

- At several critical points, the study relies on assumptions and fails to provide direct evidence supporting the suggested pathogenicity of IgA-coated bacterial MVs. Most importantly, this applies to the assumption that the PEF contains such bacterial MVs, which is not directly shown. The data shown in Fig. 4 are not completely convincing in this regard, since the EM pictures show IgA-coated microvesicles without proving their bacterial origin and for the NTA approach only aggregated statistics, but no pictures are shown.
- Along the same lines, the authors seem to take it for granted that what they find in their processed colonic lavage fluid directly reflects, what is present at the mucosal surface. Couldn't the MVs be generated from bacteria during the processing? To make the concept convincing, it is mandatory to obtain direct evidence for the presence of IgA-coated bacterial MVs in the inflamed tissue from patients with UC.
- Although the DSS data show clear differences, no further molecular analyses have been performed to support that the suggested CD89-dependent mechanism is actually causative for the phenotype. E.g., this would require to measure IL-6 and IL-8 and to assess their downstream functions. Demonstrating the reversability of the phenotype by CD89 blockade would be another piece of data clearly strengthening the concept.
- Regarding the proposed CD89-dependent mechanism, the in vitro assays were performed with cell lines. With regard to epithelial cells, using intestinal organoids should be a realistic model and with regard to monocytic cells, I would expect that this should also be possible with primary human cells.

Reviewer #2

(Remarks to the Author)

Review of "Immunoglobulin A coated bacterial membrane vesicles drive pro-inflammatory responses in ulcerative colitis":
The paper presented by Thapa et al. presents a compelling study on the contribution of immunoglobulin A coated bacterial membrane vesicles in the pathogenesis of ulcerative colitis.

Using a novel approach, the study addresses an intriguing concept regarding the role of bacterial membrane vesicles (MV) and host-derived IgA in driving intestinal inflammation in ulcerative colitis (UC). The authors isolated soluble and pelletable fractions from colonic fluids of UC patients to investigate the impact of both fractions on mucosal immune cell function. They identified that within pelletable fractions of UC patients, extracellular vesicles (EVs), in particular bacterial-MVs that exhibit high IgA-levels associated with a strong inflammatory response on IgA-receptor-positive (CD89+) immune subsets. They further claim that just IgA coated on these EVs can trigger a strong proinflammatory response. The study therefore tries to establish an association between bacterial EVs and host-produced IgA and further connects this to CD89+ immune cells. These findings suggest a novel link between these factors and CD89+ immune cells, shedding light on potential mechanisms underlying UC pathogenesis. However, while the concept is promising, the current data presented in the study

may not be entirely convincing.

One notable concern is the lack of comprehensive controls to confirm that IgG bound to bacterial MVs is the primary driving factor behind the observed pro-inflammatory responses in CD89+ immune cells. Further investigations are needed to elucidate the molecular mechanisms underlying this phenomenon and to explore why IgG presented via bacterial MVs appears to be proinflammatory while soluble IgG does not evoke similar responses.

Additionally, the authors should address whether other factors carried on or in bacterial MVs, which may vary between patients, could contribute to the observed effects. Considering the complexity of the gut microbiome and its interaction with the host immune system, it is essential to explore potential confounding variables to strengthen the study's conclusions.

Recent studies have highlighted the presence of elevated levels of immunoglobulin-coated bacteria in the feces of inflammatory bowel disease (IBD) patients. It has been proposed that these immunoglobulin-coated bacteria play a role in driving intestinal inflammation, shedding light on a potential mechanism underlying IBD pathogenesis. The novelty of the current study lies in the identification of IgA-coated bacterial MVs as potential mediators of intestinal inflammation. While this concept is intriguing, the evidence presented in the study is not sufficiently convincing.

Specific comments

a) Previous studies have convincingly established that the metabolic, lipidomic, and metabolomic profiles of bacterial MVs can exhibit variations in the context of inflammation. In this context, the current study fails to address this important aspect, which could potentially confound the observed proinflammatory effects attributed to IgG-bound bacterial EVs. To strengthen their findings and rule out the influence of other factors carried by bacterial EVs, the authors should consider incorporating data on the metabolic, lipidomic, and metabolomic content of the EVs analyzed in their study. This would provide a more comprehensive understanding of the complex interplay between bacterial EVs and host immune responses in ulcerative colitis.

b) The authors should address the lack of detailed description regarding the isolation of EVs in their study. Without comprehensive documentation of the EV isolation process, there is a risk that the isolated fractions may contain various other components besides EVs, such as flagella or other bacterial components. These contaminants may not be detectable by NTA alone. To ensure the purity and specificity of the isolated EV fractions, it is essential to include transmission electron microscopy (TEM) analysis. TEM allows for direct visualization of EV morphology and can provide valuable insights into their purity and integrity. Moreover, according to recent guidelines published by the Journal of Extracellular Vesicles, TEM analysis is considered a crucial step in EV characterization. Similarly western-blot data should be included to confirm the absence of other bacterial/dietary contaminations. According to MISEV2024 guidelines they should report TEM, NTA and western blot markers etc. I also miss data on characterization of MVs generated for in vitro and in vivo studies.

c) Authors must provide overview TEM images (electron micrographs) of the EV samples shown in Fig.4. Please show uncropped images to allow for an evaluation of the overall gold particle distribution around the samples.

d) What was used for the normalization to determine the percentage of hEVs and bEVs?

e) The statement that the human group roughly contained bEVs and hEVs summing up to 50-80% of the total protein biomass raises questions about the composition of the remaining fraction.

f) Given the availability of more sophisticated and physiologically relevant models like organoids, the use of HT-29 cells may limit the interpretation and applicability of the study's findings. Therefore, the authors should consider employing organoid models in future experiments to enhance the clinical relevance and robustness of their results.

g) The quantification of CD68+ cells in the study is not entirely convincing. The distribution of these cells among the patients appears to be quite high, which could potentially introduce variability and uncertainty into the results. Therefore, including additional patients in the analysis would strengthen the robustness and reliability of the findings. Furthermore, the authors mention the presence of CD89+CD68dim/- cells, but it is unclear how these cells were detected and characterized.

h) The authors' observation that undifferentiated U937 cells failed to induce a robust pro-inflammatory response to bacterial MVs raises the important question regarding the potential influence of cellular differentiation on this response. To elucidate whether the observed lack of response in undifferentiated U937 cells is indeed attributed to differentiation status, the authors need to conduct additional control experiments including: differentiated U937 cells exposed to bacterial MVs in the presence of an endocytosis blocker and sCD68 as well as a factor that induced inflammation independent of CD68.

i) The authors state that mucosal inflammation is associated with IgA coating of bacterial MVs, which raises questions about the necessity of artificially introducing these vesicles during DSS-colitis experiments. If mucosal inflammation is indeed linked to the presence of IgA-coated bacterial MVs, one would expect these vesicles to already be present during DSS-colitis without the need for additional administration. Moreover, it is essential for the authors to consider whether these exogenously administered vesicles can effectively reach the gut and exert their intended effects. Given the complexities of gut physiology and the potential for vesicle clearance or degradation, it is crucial to label these vesicles to trace their biodistribution accurately.

Reviewer #3

(Remarks to the Author)

This study presents a compelling investigation showing that IgA-coated bacterial MVs elicit a proinflammatory response in a CD89-dependent manner. The findings contribute significantly to our understanding of the mechanisms underlying inflammatory responses in ulcerative colitis (UC) and offer potential insights for drug development. The study is conducted very well, employing a combination of in vitro and in vivo assays utilizing human patient samples and cell cultures, resulting in a convincing conclusion. I only have some minor comments:

1. In line 127 and elsewhere, it would be preferable to use the term "extracellular vesicles (EVs)" rather than "MV," for the human EV and bacterial MV mixtures, as "MV" typically refers to bacterial EVs, while EVs refer to a broader range of

vesicles.

2. In lines 143 to 144, please clarify what "%" refers to. It seems to indicate the percentage relative to the total protein biomass.

3. In line 154, is there a specific reason for using only one bacterial species in the LTA assay compared to two bacteria in the LPS assay for the standard? While not critical, explaining any rationale behind this choice would be beneficial.

4. In Fig. 1a, is there a representative photo of non-IBD included? It is unclear what the blue box indicates for non-IBD.

5. In Figs. 2 and 3, the MV fraction was dissolved in saline. Was a mock control (saline only) used for the MV fractions in the assays shown in Figs. 2 and 3?

6. In line 272, please provide further explanation for why ETEC MVs were used.

7. In Fig. 5, while I understand the rationale for using a mixture of *B. fragilis* and *L. acidophilus* MVs, are there any available data on their individual use? Including such information may provide valuable insights into whether the MVs have synergistic effects or not.

8. Typically, bacterial flagella are detected in MV fraction unless further purified. Therefore, stating that *B. fragilis* and *L. acidophilus* are non-flagellated bacteria, and at least MV fractions from these bacteria are devoid of flagella, could be important. In relation to this, were any flagella observed in the PEF fraction?

9. Figs. 5c and d. It is shown that while uncoated MVs stimulated IL-6 and 8 release, they do not further stimulate cytokine production with increased dosage. In contrast, IgA-coated MVs induce a dose-dependent cytokine response. It may be worth discussing why such a difference in the dose-dependent manner occurs.

10. Regarding line 395, could there be further discussion on whether IgA-coated MVs or IgA-coated bacterial cells contribute to a larger extent to the proinflammatory response in UC?

11. Line 547. It would be helpful to provide a reference or additional information that the vesicles are lysed with 0.1% SDS.

Version 1:

Reviewer comments:

Reviewer #1

(Remarks to the Author)

The authors have done a good job in addressing the reviewers' comments. A few concerns could not be solved with new experimental data, but these aspects have been adequately discussed in the revised version of the manuscript.

Reviewer #3

(Remarks to the Author)

The authors did a great job in improving the manuscript. My concerns have been cleared and do not have further comments.

REVIEWER COMMENTS

Reviewer #1 (Remarks to the Author):

In this manuscript, Thapa et al. report on their studies on IgA-coated bacterial microvesicles in the context of and as a driver of ulcerative colitis. The authors suggest that IgA-coated bacterial microvesicles are increased in active UC and have an important pro-inflammatory function by activating CD89+ immune cells to produce cytokines.

Although this is conceptually appealing and might help to explain previously unrecognized aspects of microbiota-driven IBD pathogenesis, several aspects limit the significance of the findings.

Major points:

- At several critical points, the study relies on assumptions and fails to provide direct evidence supporting the suggested pathogenicity of IgA-coated bacterial MVs. Most importantly, this applies to the assumption that the PEF contains such bacterial MVs, which is not directly shown. The data shown in Fig. 4 are not completely convincing in this regard, since the EM pictures show IgA-coated microvesicles without proving their bacterial origin and for the NTA approach only aggregated statistics, but no pictures are shown.

Author response:

We thank the reviewer for the critical evaluation. To address the reviewer's comment, we performed immunogold co-staining to detect human IgA (20 nm gold particles) and bacterial vesicle markers (i.e. LTA or OmpA, 6 nm gold particles) on the same vesicles in samples from active UC patients and IgA-coated bacterial extracellular vesicles (BEVs) derived from *L. acidophilus* and *B. fragilis*. We show representative images of co-labelled vesicles from these samples in Fig. 4a. In contrast, uncoated bacterial vesicles derived from *L. acidophilus* and *B. fragilis* are only labeled with 6 nm gold particles (bacterial markers), but not with 20 nm gold particles (human IgA). Thus, uncoated BEVs served as negative control indicating specific binding of the 20 nm gold particle-conjugated anti-human IgA-antibody. This data is provided in the new Fig. 4a in the revised version of manuscript.

In addition, we performed immunoprecipitation experiments with PEF samples derived from three active UC patients using magnetic beads coated with anti-human IgA antibodies. Our results demonstrate enrichment of LTA-containing nanoparticles by fluorescent NTA as well as enrichment of bacterial surface proteins by MS analyses of pulldown samples compared to the original PEFs for all three patient samples analyzed. The data is provided in the new Fig. 4c and 4d. The enrichment of these bacterial vesicle biomarkers in the pulldown fraction provide strong evidence that human IgA is bound to particles of bacterial origin.

Moreover, the reviewer might have missed the sandwich ELISA data (Fig. 5a and 5b in the original version) presented now in the new Fig. 4e. This analysis was designed to detect immunocomplexes of human IgA bound to bona fide bacterial vesicle markers (i.e. LTA for Gram-positive origin and OmpA for Gram-negative origin) in the PEF samples. In detail, the ELISA was based on anti-LTA or anti-OmpA (targeting bacterial vesicle marker of Gram positive and negative bacteria) as coating antibodies to capture BEVs in the EV fractions, followed by HRP-conjugated anti-human IgA as detection antibody to quantify the IgA bound to BEVs. The data shows significantly higher amounts of such immunocomplexes in active UC patients samples compared to the non-IBD group.

Regarding visualization images obtained by NTA, please note that the technology relies on the illumination of a liquid sample by a laser beam and the light scattered by particles moving under 3-dimensional Brownian motion is recorded as a video using a camera. As described in the method section, we recorded ten 60 sec videos for each sample in the camera level 16 and analyzed in the batch-processing mode. Since NTA is based on the three-dimensional movement of particles in a liquid suspension the generated images are typically of insufficient resolution to visualize morphological features. We believe that the TEM images provided in the revised version are of much better quality demonstrating i) presence of vesicles (Suppl. Fig. S2) and ii) immunogold co-staining of vesicles detecting human IgA and a bacterial vesicle marker (i.e. LTA or OmpA) on the same vesicle (Fig. 4a).

In summary, the data sets provided in the new Fig. 4 including the sandwich ELISA and NTA data already present in the original version as well as the additional data sets (enrichment of bacterial markers by immunoprecipitation and immunogold co-staining) generated during revision underpin the presence of IgA-coated bacterial vesicles in the PEF samples of active UC patients.

- Along the same lines, the authors seem to take it for granted that what they find in their processed

colonic lavage fluid directly reflects, what is present at the mucosal surface. Couldn't the MVs be generated from bacteria during the processing? To make the concept convincing, it is mandatory to obtain direct evidence for the presence of IgA-coated bacterial MVs in the inflamed tissue from patients with UC.

Author response:

We value the comment of the reviewer. Changes during processing essentially apply to all samples collected from human cohorts as well as to the documented isolations of bacterial-derived extracellular vesicles (BEVs) from the gastrointestinal tract reported in recent studies (e.g. doi: 10.1128/aem.00533-22; DOI: 10.1038/s41596-019-0236-5; 10.3389/fcimb.2021.667987, 10.1371/journal.pone.0076520).

We took several measures to minimize this possibility: Highly efficient communication between clinicians and lab scientists enabled samples obtained from patients to be processed within 1 h. Rapid turnover was made easier by specifically selecting colonic luminal fluid as starting material as these samples are typically cleaner than stool.

We state the reasons for this preference as follows: “Stool harbors substantial amounts of indigestible material and has variable quality due to individual diet and unpredictable turn-around times until processing. In contrast colonic luminal fluid is cleaner due the patient’s bowel preparation and fasting prior to colonoscopy and in practice, can be directly processed. Furthermore, mucosal washing during colonoscopy also collects microbial samples that are attached to the colonic mucosal surface¹².”

Moreover, samples were kept on ice to slow down degradation and metabolic activity. Thus, we think the optimized isolation protocol avoids potential artefacts generated during the processing. Because of these measures we think that the abundance of vesicles produced ex vivo is relatively low, but of course we cannot entirely exclude that possibility. In the manuscript we now state: “Although the samples were processed within 1 h and kept on ice to minimize metabolic activity, we cannot completely rule out degradation or modifications in the colonic fluid during this time period.”

With regard to direct visualization of IgA-coated bacterial vesicles in inflamed tissue from patients, we agree that this would be highly valuable. However, IBD in general and UC specifically is characterized by ulceration of the mucosal layer (i.e. a barrier breach) thus and this has been shown in multiple studies, luminal contents can reach immune cells present in the tissue (e.g. doi: 10.1038/nrgastro.2015.18).

To address the visualization, the BEV size will require electron microscopy of human biopsies as light microscopes cannot currently provide sufficient resolution of such nanoparticles. Thus, we are limited by the currently available tools and technologies. We would like to emphasize that we view BEVs as non-living facsimiles of the bacterial cells. Whether IgA-coated bacteria or BEVs are more important in exacerbating inflammation remains to be elucidated, but it is likely that both can be contributing factors. Current studies aiming to detect bacterial vesicles in human tissue essentially only show presence of a bacterial surface marker by fluorescence microscopy, but fail to provide sufficient resolution to visualize vesicle morphology. An approach to allow direct visualization of IgA-coated bacterial vesicles will likely require a screen of hundreds of ultrathin tissue sections prepared for electron microscopy followed by immunogold labeling to ensure bacterial origin and coating with human IgA. Even if a co-stained vesicle-shaped structure can be detected via this technique, it cannot be ruled out that it represents a bacterial cell cut close to its surface, which would appear as a circular structure enclosed by a membrane similar to a vesicle. In general patients’ biopsies are routinely fixed in formalin, which disintegrates tissue architecture to a certain extend. Changing the fixation process would need an amendment of the current practice of biopsy taking in IBD patients. Thus, the visualization of IgA-coated BEVs in human biopsies is impossible with the currently available techniques and instruments. We added these thoughts to the discussion as follows:

“Under homeostatic conditions IgA-opsonized bacteria are primarily restricted to the intestinal lumen, while under pathological conditions they can reach the lamina propria to interact with the immune cells^{59,60}. Similar principles may apply to BEVs, but whether IgA-coated bacteria or BEVs are more important in exacerbating inflammation remains to be elucidated. The direct visualization of BEVs and their differentiation from bacterial cells in the human intestine is limited by the available methods, as current light microscopy cannot visualize vesicle with a sufficiently high resolution. However, BEVs are non-living facsimiles of the donor cell vesicles and therefore share the same epitopes as the bacterial cell. Mechanistic studies on how BEVs spread in vivo are limited. However, it is believed that BEVs can pass the intestinal barrier more easily than bacteria and can deliver a concentrated dose of pro-inflammatory cargo, e.g. LPS, LTA, surface proteins or peptidoglycan^{15,61}.”

- Although the DSS data show clear differences, no further molecular analyses have been performed to support that the suggested CD89-dependent mechanism is actually causative for the phenotype.

E.g., this would require to measure IL-6 and IL-8 and to assess their downstream functions. Demonstrating the reversability of the phenotype by CD89 blockade would be another piece of data clearly strengthening the concept.

Author response:

WT mice do not have a CD89 homolog (doi: 10.1073/pnas.94.10.5261; doi: 10.1074/jbc.272.11.7320). Thus, the DSS data was generated using transgenic mice expressing the human CD89 on myeloid cells (CD89) and wildtype littermates (LM). Strong inflammation highlighted by histology and lipocalin levels was only observed in transgenic mice expressing the CD89 receptor in combination with the administration of human IgA-coated bacterial extracellular vesicles (BEVs), but not in wildtype littermates receiving the same treatment. As CD89 and LM mice only differ by the presence or absence of CD89, this result clearly pinpoints the mechanism to be CD89-dependent.

We also provide IL-6 and TNF-alpha cytokine responses of bone marrow derived cells (BMDCs) from CD89+ and LM mice upon exposure to uncoated and IgA-coated BEVs. Again, only CD89+ murine cells exposed to IgA-coated BEVs showed a pronounced pro-inflammatory cytokine response. Since mice lack IL-8, this analysis is not possible. Instead we chose lipocalin for in vivo assessment as this is a clinically relevant, genuine biomarker for intestinal inflammation in UC (doi: 10.1093/ecco-jcc/jjaa124). In vivo assessed cytokine levels like IL-6 are not specific for UC or IBD, but can be elevated in various inflammatory diseases and infections. Therefore, measurements of cytokines are currently not recommended as a valid biomarkers in the clinical management of IBD (doi: 10.1053/j.gastro.2022.12.007).

To characterize downstream functions and test inhibition of the CD89-dependent pathway, we performed cell culture experiments using the commercially available Syk-inhibitor R406 (InvivoGen). As demonstrated previously, Syk represents an essential downstream factor in the CD89-dependent signaling pathway in CD89 mice (doi: 10.1038/s41385-019-0167-z.). Thus, we used the Syk-inhibitor R406 to assess the downstream function and block the CD89-pathway. Indeed, presence of the Syk-inhibitor significantly reduced the IL-6 and TNF-alpha cytokine responses evoked by IgA-coated BEVs or by EV samples from active UC patients. The new data is provided in Fig. 6 and discussed in the appropriate sections of the manuscript as follows:

“As demonstrated recently, the tyrosine kinase Syk represents an essential downstream factor in the CD89-dependent signaling pathway in CD89 mice⁵¹. Treatment of CD89+ BMDCs with the Syk-inhibitor R406 significantly decreased IgA-coated BEVs^{B/L} mediated IL-6 and TNF-alpha responses (Fig. 6b). BMDC isolated from LM and CD89 mice are ideally suited to demonstrate CD89-dependent effects as they only differ in the expression of human CD89. Thus, we also assessed their cytokine responses upon exposure to seven randomly selected representative EV fractions from active UC patients and non-IBD controls (Fig. 6c). In BMDCs isolated from LM the EV fractions from both groups, i.e. active UC patients and non-IBD controls, induced similar, low level cytokine responses. Consistent with observations made with the CD89+CD14+CD11b+ U937 monocytes and primary human monocytes, EV fractions from active UC patients induced significantly higher IL-6 and TNF-alpha responses in CD89+ BMDCs compared to non-IBD controls. Treatment of CD89+ BMDCs with the Syk-inhibitor R406 significantly decreased IL-6 and TNF-alpha responses mediated by EV fractions from three randomly selected active UC patients (Fig. 6d). These results underpin the importance of CD89 including its downstream pathways for the strong pro-inflammatory responses evoked by IgA-coated BEVs^{B/L} as well as by the EV fractions from active UC patients.”

- Regarding the proposed CD89-dependent mechanism, the in vitro assays were performed with cell lines. With regard to epithelial cells, using intestinal organoids should be a realistic model and with regard to monocytic cells, I would expect that this should also be possible with primary human cells.

Author response:

We agree and added primary cell data, i.e. human intestinal organoids as well as human primary monocytes. The new data sets are provided in new Fig. 3 and 5 and include:

- FACS analyses confirming that human intestinal organoids do not express CD89, whereas primary human monocytes show CD89 expression. We kindly refer to new Fig. 3a and 3b.
- The cytokine response (human IL-8) of human intestinal organoids from two donors after exposure to EV samples from non-IBD and active UC patients. No significant differences in cytokine levels between exposure of non-IBD and active UC patients samples were detected. We kindly refer to new Fig. 3c.
- The cytokine response (human IL-8) in human intestinal organoids from two donors after exposure to IgA-coated and uncoated bacterial extracellular vesicles from *L. acidophilus* and *B. fragilis*

(BEV^{B/L}). No significant differences in cytokine levels were detected between those exposed to IgA-coated or uncoated BEV^{B/L}. We kindly refer to new Fig. 5c.

- The cytokine responses (human IL-8 & IL-6) in primary human monocytes obtained from three independent donors after exposure to EV samples from non-IBD and active UC patients. For all three donors, significantly higher cytokine levels were detected after exposure to active UC patients samples compared to non-IBD samples. We kindly refer to new Fig. 3d.
- The cytokine responses (human IL-8 & IL-6) in primary human monocytes obtained from three independent donors after exposure to IgA-coated and uncoated bacterial extracellular vesicles from *L. acidophilus* and *B. fragilis* (BEV^{B/L}). For all three donors, significantly higher cytokine levels were detected after exposure to IgA-coated BEV^{B/L} compared to uncoated BEV^{B/L}. We kindly refer to new Fig. 5d.

In summary, the new data on human intestinal organoids and primary human monocytes mirror the results obtained with the continuously passaged cell lines (HT-29 and CD89+CD14+CD11b+ U937 monocytes), which strengthens the CD89-dependent mechanism identified in this study. It is important to note, that the yield of primary cells is limited. Unlike HT-29 and U937, which can be continuously passaged and are therefore optimally suited for screens, the primary cell quantity and quality varies between donors. Thus, the experiments with primary cells were restricted to a limited number of independent samples per donor. Moreover, donor-specific variation in the cytokine responses is high. Thus, while these important complementary approaches have added data that strengthen our findings, they cannot replace the primary screen with immortalized cells, which can be continuously passaged.

We discuss the data in the appropriate sections of the manuscript:

“We confirmed the results obtained with CD89-negative HT-29 and CD89+CD14+CD11b+ U937 monocytes, in more physiologically relevant cell systems, i.e. human intestinal organoids and primary human monocytes (Fig. 3). The CD89 expression profiles analyzed by FACS verified that human intestinal organoids are CD89-negative, while primary human monocytes are CD89-positive (Fig. 3a and b). Unlike HT-29 and U937, which can be continuously passaged and are therefore optimally suited for screens, the primary cell quantity and quality varies between donors. Thus, we randomly selected five to eight representative EV fractions from the active UC patients and non-IBD controls for these assays. EV fractions of active UC patients and non-IBD controls induced similar IL-8 responses in both batches of human intestinal organoids tested (Fig. 3c). In contrast, EV fractions from UC patients consistently induced significantly higher levels of IL-6 and IL-8 in primary human monocytes from three different donors compared to EV fractions from non-IBD controls (Fig. 3d).”

and

“Similar to the EV fraction experiments, we also exposed uncoated BEVs^{B/L} and IgA-coated BEVs^{B/L} to CD89-negative human intestinal organoids as well as CD89-positive primary human monocytes to confirm the cytokine responses in more physiologically relevant cell systems (Fig. 5c and d). In line with previous results, primary IgA-coated BEVs^{B/L} induced significantly higher pro-inflammatory cytokine responses in human monocytes from three different donors compared to uncoated BEVs^{B/L}, but not in human intestinal organoids.”

Reviewer #2 (Remarks to the Author):

Review of "Immunoglobulin A coated bacterial membrane vesicles drive pro-inflammatory responses in ulcerative colitis":

The paper presented by Thapa et al. presents a compelling study on the contribution of immunoglobulin A coated bacterial membrane vesicles in the pathogenesis of ulcerative colitis. Using a novel approach, the study addresses an intriguing concept regarding the role of bacterial membrane vesicles (MVs) and host-derived IgA in driving intestinal inflammation in ulcerative colitis (UC). The authors isolated soluble and pelletable fractions from colonic fluids of UC patients to investigate the impact of both fractions on mucosal immune cell function. They identified that within pelletable fractions of UC patients, extracellular vesicles (EVs), in particular bacterial-MVs that exhibit high IgA-levels associated with a strong inflammatory response on IgA-receptor-positive (CD89+) immune subsets. They further claim that just IgA coated on these EVs can trigger a strong proinflammatory response. The study therefore tries to establish an association between bacterial EVs and host-produced IgA and further connects this to CD89+ immune cells. These findings suggest a novel link between these factors and CD89+ immune cells, shedding light on potential mechanisms underlying UC pathogenesis. However, while the concept is promising, the current data presented in the study may not be entirely convincing.

One notable concern is the lack of comprehensive controls to confirm that IgG bound to bacterial MVs is the primary driving factor behind the observed pro-inflammatory responses in CD89+ immune cells. Further investigations are needed to elucidate the molecular mechanisms underlying this phenomenon and to explore why IgG presented via bacterial MVs appears to be proinflammatory while soluble IgG does not evoke similar responses.

Additionally, the authors should address whether other factors carried on or in bacterial MVs, which may vary between patients, could contribute to the observed effects. Considering the complexity of the gut microbiome and its interaction with the host immune system, it is essential to explore potential confounding variables to strengthen the study's conclusions.

Recent studies have highlighted the presence of elevated levels of immunoglobulin-coated bacteria in the feces of inflammatory bowel disease (IBD) patients. It has been proposed that these immunoglobulin-coated bacteria play a role in driving intestinal inflammation, shedding light on a potential mechanism underlying IBD pathogenesis. The novelty of the current study lies in the identification of IgA-coated bacterial MVs as potential mediators of intestinal inflammation. While this concept is intriguing, the evidence presented in the study is not sufficiently convincing.

Specific comments

a) Previous studies have convincingly established that the metabolic, lipidomic, and metabolomic profiles of bacterial MVs can exhibit variations in the context of inflammation. In this context, the current study fails to address this important aspect, which could potentially confound the observed proinflammatory effects attributed to IgG-bound bacterial EVs. To strengthen their findings and rule out the influence of other factors carried by bacterial EVs, the authors should consider incorporating data on the metabolic, lipidomic, and metabolomic content of the EVs analyzed in their study. This would provide a more comprehensive understanding of the complex interplay between bacterial EVs and host immune responses in ulcerative colitis.

Author response:

We agree with the reviewer that metabolic, lipidomic and metabolomic analyses of patient samples in defined cohorts can be an approach to identify markers correlating with the disease. In contrast, our study was designed to interrogate the pro-inflammatory potency in fractions of colonic fluid samples. As much as we value the reviewer's idea, the yield in the EV fractions does not allow both, the comprehensive characterization of pro-inflammatory potency as well as an elaborative OMIC-based untargeted approach.

An OMIC-based untargeted approach would require many more patients and better defined subgroups with regard to disease severity, disease history, diet, sex, medication, alcohol and tobacco consumption, etc. The patient cohort studied here is too heterogenous and too small to identify clear correlations based on OMICs-analyses. This assessment is based on a recent study that analyzed 154 stool samples of patients with irritable bowel syndrome, UC and controls for microbiome composition via amplicon sequencing and metabolites via untargeted metabolomics (doi.org/10.1080/19490976.2024.2359500). Although this study included twice as many patients as ours, no clear correlation of a defined effector class with UC could be found. The authors of this study conclude that "...a drawback of this study is the exploratory scope with a limited sample size and complex disease populations. Particularly in the UC group, a larger sample size to compare clinical features (disease activity, location, therapy) could have increased the outcome. In addition, more

detailed medication history including more specific timelines on antibiotic, probiotic and proton pump inhibitors intake could have been useful in interpretation.”

In contrast to such an OMIC-based study we took a targeted approach to trace pro-inflammatory effectors present in UC patients by filtration and centrifugation steps. Our findings allowed us to identify IgA-coated bacterial vesicles as a driver of inflammation and to link the inflammatory response to the human IgA-receptor CD89. We thereby provide new mechanistic insights underlying the disease instead of identifying causative or correlative changes in the microbiome, lipidome or metabolome. Having identified IgA-coated vesicles as a driver for inflammation, we indeed already work on a more defined OMIC-based follow up study to identify mucosal immunoglobulin epitope repertoire from UC patients. We think that with the existing patient cohort our strategy is more efficient as we now use the full power of a targeted OMIC-analyses focusing on Ig epitope repertoire in human samples.

In summary the suggested metabolic, lipidomic and metabolomic analyses of the patient samples represents an independent study, that in its magnitude, is beyond the scope of this manuscript.

b) The authors should address the lack of detailed description regarding the isolation of EVs in their study. Without comprehensive documentation of the EV isolation process, there is a risk that the isolated fractions may contain various other components besides EVs, such as flagella or other bacterial components. These contaminants may not be detectable by NTA alone. To ensure the purity and specificity of the isolated EV fractions, it is essential to include transmission electron microscopy (TEM) analysis. TEM allows for direct visualization of EV morphology and can provide valuable insights into their purity and integrity. Moreover, according to recent guidelines published by the Journal of Extracellular Vesicles, TEM analysis is considered a crucial step in EV characterization. Similarly western-blot data should be included to confirm the absence of other bacterial/dietary contaminations. According to MISEV2024 guidelines they should report TEM, NTA and western blot markers etc. I also miss data on characterization of MVs generated for in vitro and in vivo studies.

Author response:

We agree with the reviewer and added further characterization of the EV fractions including TEM. However, we would like to point out that several data sets characterizing the EV fractions were already provided in the original manuscript. Maybe the reviewer missed Fig, 1 c - f, Table 1 (entitled “Qualitative and quantitative analyses of the isolated EVs used in this study”) and supplementary Figure S3 (entitled “Characterization of the pelletable fraction composition”). These data sets were already present in the original version. Specifically, this included NTA analyses (diameter and particle amount), protein biomass quantification, LTA and LAL assays to quantify bona-fide biomarkers of bacterial extracellular vesicles as well as dot blot analyses (same principle as immunoblot, but allowing high-throughput analyses) of CD9 and CD81 representing bona fide host-derived extracellular vesicle markers.

To enhance this set we added the suggested TEM morphology and substantially extended the EV characterization including several data sets in accordance with MISEV 2023. The revised version includes:

- TEM images of vesicles in the EV fraction from human colonic fluid samples derived from non-IBD and active UC group. We kindly refer to new Fig. S2
- TEM images of vesicles formed in vitro including host-derived extracellular vesicles from HT-29 and bacterial-derived vesicles from *L. acidophilus* and *B. fragilis*. We kindly refer to new Fig. S2.
- immunogold co-staining to detect human IgA and a bacterial vesicle marker (i.e. LTA or OmpA) on the same vesicle in samples from active UC patients and IgA-coated bacterial-derived extracellular vesicles derived from *L. acidophilus* and *B. fragilis*. We kindly refer to new Figure 4a.
- protein biomass quantification of EV fractions from human colonic fluid samples and bacterial-derived extracellular vesicles. We kindly refer to new Fig. 1d and Table 1.
- quantitative dot blot analyses of the EV fractions from human colonic fluid samples for three bona fide host-derived extracellular vesicle markers (i.e. CD9, CD81 and CD63 – all three recommended in MISEV2023). We kindly refer to new Suppl. Fig. S3e to h.
- quantitative ELISA analyses of LTA (biomarker for Gram-positive vesicles) of EV fractions from human colonic fluid samples and bacterial-derived extracellular vesicles. We kindly refer to new Suppl. Fig. S3c and Table 1.
- quantitative analyses of LPS (biomarker for Gram-negative vesicles) by a commercially available endotoxin quantification kit (LAL) of EV fractions from human colonic fluid samples and bacterial-derived extracellular vesicles. We kindly refer to new Suppl. Fig. S3d and Table 1.

- NTA analyses incl. size and particle amounts of EV fractions from human colonic fluid samples and bacterial-derived extracellular vesicles. We kindly refer to Fig. 1e-f and Table 1.
- SYTO-9 and FM64-4 assays to assess the nucleic acid and lipid amounts of EV fractions from human colonic fluid samples and bacterial-derived extracellular vesicles. We kindly refer to new Suppl. Fig. S3a-b and Table 1.

Our samples and the isolation process are described in detail in the Method sections entitled “Collection of colonic luminal fluid and separation into soluble and pelletable (MV) effector fractions” as well as “Bacterial strains, growth conditions and bacterial MV isolation”, which provide a fair and detailed description of every step in the isolation process.

We would also like to briefly comment on the MISEV reference mentioned by the reviewer. While the authors acknowledge the value of the MISEV 2023 initiative (doi: 10.1002/jev2.12404), it is important to note that these guidelines focus primarily on eukaryotic EVs (e.g. exosomes). Colonic fluid samples as a source of bacterial-derived EVs is not covered by MISEV 2023. For that reason, the corresponding author is an active member of the “ISEV Task Force for Bacterial Extracellular Vesicles (BEVs)”. The aim is to establish standard markers and purification methods for bacterial-derived vesicles which will need to differ from the current MISEV 2023 guidelines. For example, based on existing data the current task force concluded that BEV isolation with a single “pelleting” step is perfectly fine for some species or samples, while BEV preparations from other species could benefit from a two-step process. In fact, some task force members even believe that additional “purification” steps can influence BEV make-up and mask BEV activities. Therefore, studies with samples containing EVs from different species are a major challenge and the current guidelines do not sufficiently cover characterization parameters for such sample sets. Despite these ongoing developments, the “ISEV Task Force for Bacterial Extracellular Vesicles (BEVs)” agrees that detailed methodology descriptions are essential. Thus, we described the isolation of the human EV fraction and BEV isolation in great detail. However, in line with the MISEV initiative the BEV task force also believes that prescriptive rules are not appropriate, and guidelines are guidelines not rules. MISEV 2023 (doi: 10.1002/jev2.12404) clearly states that the recommendations provided in the manuscript are not meant to prevent publication or funding of a particular project.

c) Authors must provide overview TEM images (electron micrographs) of the EV samples shown in Fig.4. Please show uncropped images to allow for an evaluation of the overall gold particle distribution around the samples.

Author response:

We agree and provided uncropped as well as enlarged images in the revised version: Due to a comment by reviewer 1 we repeated the TEMs with two differentially-sized immunogold conjugated antibody system allowing co-staining to detect human IgA (20 nm gold particles) and a bacterial vesicle marker (i.e. LTA or OmpA, 6 nm gold particles) on the vesicles. Please find the new images in Fig. 4a. Moreover, the new Suppl. Fig. S2 shows TEM images of vesicles in negatively stained EV fractions.

d) What was used for the normalization to determine the percentage of hEVs and bMVs?

Author response:

Please refer to the appropriate result paragraph (page 6) explaining that vesicles derived from the intestinal epithelial cell line HT-29 were used as standard to determine the percentage of hEVs. For clarification and better understanding of the quantification we provide representative dot blots for all three biomarkers in the revised version and extended the figure legends (new Suppl. Fig. S3e to h) as well as method sections (entitled “Dot-blot”). For LTA the bacterial extracellular vesicles from *L. acidophilus* were used as standard. For LPS reactogenicity (LAL assay) we refer to the bacterial extracellular vesicles from *B. fragilis* and ETEC as standard. All samples were normalized to protein biomass as indicated in the figure legend of Suppl. Figure S3.

e) The statement that the human group roughly contained bMVs and hEVs summing up to 50-80% of the total protein biomass raises questions about the composition of the remaining fraction.

Author response:

We can only speculate on the composition of the remaining fraction. As stated in the original and revised version of the manuscript the quantification can only be a rough approximation. Moreover, the detected distribution in EV fractions including not closer definable matter is similar to a recently reported state-of-the-art protocol for MV isolation from human body fluids. We added the following statement:

“It should be noted, however, that this is only a rough approximation as (i) the LTA antibody might not detect all Gram-positive BEV species, (ii) LPS derived from diverse species shows different reactogenicity in LAL assays, and (iii) bacterial explosive membrane vesicles enriched with cytoplasmic content may not be adequately detected by these assays^{15,26}.”

and

“The distribution in the PEF samples is similar to a recently reported state-of-the-art protocol for EV isolation from human stool²⁸.”

We added a statement on the potential composition of the remaining fraction:

“In addition to undetectable microbial vesicles, the remaining fraction could contain undigested food, intestinal mucus, fibers, flagella and pili. However, TEM analyses of human-derived EV fractions showed no obvious bacterial flagellar- and pili-like structures (Fig. S2). The distribution in the PEF samples is similar to a recently reported state-of-the-art protocol for EV isolation from human stool²⁸.”

f) Given the availability of more sophisticated and physiologically relevant models like organoids, the use of HT-29 cells may limit the interpretation and applicability of the study's findings. Therefore, the authors should consider employing organoid models in future experiments to enhance the clinical relevance and robustness of their results.

Author response:

We agree and added primary cell data, i.e. human intestinal organoids as well as human primary monocytes. The new data sets are provided in new Fig. 3 and 5 and include:

- FACS analyses confirming that human intestinal organoids do not express CD89, whereas primary human monocytes show CD89 expression. We kindly refer to new Fig. 3a and 3b.
- The cytokine response (human IL-8) of human intestinal organoids from two donors after exposure to EV samples from non-IBD and active UC patients. No significant differences in cytokine levels between exposure of non-IBD and active UC patients samples were detected. We kindly refer to new Fig. 3c.
- The cytokine response (human IL-8) in human intestinal organoids from two donors after exposure to IgA-coated and uncoated bacterial extracellular vesicles from *L. acidophilus* and *B. fragilis* (BEV^{B/L}). No significant differences in cytokine levels were detected between those exposed to IgA-coated or uncoated BEV^{B/L}. We kindly refer to new Fig. 5c.
- The cytokine responses (human IL-8 & IL-6) in primary human monocytes obtained from three independent donors after exposure to EV samples from non-IBD and active UC patients. For all three donors, significantly higher cytokine levels were detected after exposure to active UC patients samples compared to non-IBD samples. We kindly refer to new Fig. 3d.
- The cytokine responses (human IL-8 & IL-6) in primary human monocytes obtained from three independent donors after exposure to IgA-coated and uncoated bacterial extracellular vesicles from *L. acidophilus* and *B. fragilis* (BEV^{B/L}). For all three donors, significantly higher cytokine levels were detected after exposure to IgA-coated BEV^{B/L} compared to uncoated BEV^{B/L}. We kindly refer to new Fig. 5d.

In summary, the new data on human intestinal organoids and primary human monocytes mirror the results obtained with the continuously passaged cell lines (HT-29 and CD89+CD14+CD11b+ U937 monocytes), which strengthens the CD89-dependent mechanism identified in this study. It is important to note, that the yield of primary cells is limited. Unlike HT-29 and U937, which can be continuously passaged and are therefore optimally suited for screens, the primary cell quantity and quality varies between donors. Thus, the experiments with primary cells were restricted to a limited number of independent samples per donor. Moreover, donor-specific variation in the cytokine responses is high. Thus, while these important complementary approaches have added data that strengthen our findings, they cannot replace the primary screen with immortalized cells, which can be continuously passaged.

We discuss the data in the appropriate sections of the manuscript:

“We confirmed the results obtained with CD89-negative HT-29 and CD89+CD14+CD11b+ U937 monocytes, in more physiologically relevant cell systems, i.e. human intestinal organoids and primary human

monocytes (Fig. 3). The CD89 expression profiles analyzed by FACS verified that human intestinal organoids are CD89-negative, while primary human monocytes are CD89-positive (Fig. 3a and b). Unlike HT-29 and U937, which can be continuously passaged and are therefore optimally suited for screens, the primary cell quantity and quality varies between donors. Thus, we randomly selected five to eight representative EV fractions from the active UC patients and non-IBD controls for these assays. EV fractions of active UC patients and non-IBD controls induced similar IL-8 responses in both batches of human intestinal organoids tested (Fig. 3c). In contrast, EV fractions from UC patients consistently induced significantly higher levels of IL-6 and IL-8 in primary human monocytes from three different donors compared to EV fractions from non-IBD controls (Fig. 3d).”

and

“Similar to the EV fraction experiments, we also exposed uncoated BEVs^{B/L} and IgA-coated BEVs^{B/L} to CD89-negative human intestinal organoids as well as CD89-positive primary human monocytes to confirm the cytokine responses in more physiologically relevant cell systems (Fig. 5c and d). In line with previous results, primary IgA-coated BEVs^{B/L} induced significantly higher pro-inflammatory cytokine responses in human monocytes from three different donors compared to uncoated BEVs^{B/L}, but not in human intestinal organoids.”

g) The quantification of CD68+ cells in the study is not entirely convincing. The distribution of these cells among the patients appears to be quite high, which could potentially introduce variability and uncertainty into the results. Therefore, including additional patients in the analysis would strengthen the robustness and reliability of the findings. Furthermore, the authors mention the presence of CD89+CD68^{dim/-} cells, but it is unclear how these cells were detected and characterized.

Author response:

We agree with the reviewer and added additional biopsies from incoming patients. In the revised version the numbers for each group are doubled. We also distinguished between CD89+/CD68+ cells (monocyte-derived cells) and CD89+CD68^{dim/-} cells (neutrophilic granulocytes) in the quantification. Both CD89+ populations are significantly increased in acute UC patients compared to non-IBD controls. We state: “In agreement with previous reports ⁴⁴, immunofluorescence imaging of human colonoscopy biopsies showed an increase of CD89+ cells in UC patients (Fig. 2e). Comprehensive quantification revealed significantly increased numbers of CD89+CD68+ cells matching the characteristics of monocyte-derived cells as well as CD89+CD68^{dim/-} cells displaying neutrophilic granulocytes in active UC patients compared to the non-IBD controls.”

h) The authors' observation that undifferentiated U937 cells failed to induce a robust pro-inflammatory response to bacterial MVs raises the important question regarding the potential influence of cellular differentiation on this response. To elucidate whether the observed lack of response in undifferentiated U937 cells is indeed attributed to differentiation status, the authors need to conduct additional control experiments including:
differentiated U937 cells exposed to bacterial MVs in the presence of an endocytosis blocker and sCD68 as well as a factor that induced inflammation independent of CD68.

Author response:

We agree with the reviewer and added substantially more data sets in the revised version. First we assessed the cytokine response in undifferentiated U937 cells and CD89+CD14+CD11b+ U937 monocytes exposed to purified bacterial peptidoglycan (PGN), which is recognized by toll-like receptors (add ref DOI: 10.1074/jbc.M107057200; DOI: 10.1074/jbc.274.25.17406). PGN mediated a significant induction of IL-8 levels in CD89+CD14+CD11b+ U937 monocytes as well as undifferentiated U937 cells (Fig. S4d). The result is also confirmed by the exposure of undifferentiated U937 cells to ETEC-derived BEVs (Fig. S4f), which also induces an IL-8 response in undifferentiated U937 cells. Although IL-8 levels in undifferentiated U937 were generally lower than in CD89+CD14+CD11b+ U937 monocytes, this result demonstrates that undifferentiated U937 are in principle capable of triggering a cytokine response upon exposure to bacterial effectors.

Please note that CD68 is not significant for this study which focuses on CD89. The differentiation of U937 to CD89+CD14+CD11b+ U937 monocytes could also alter other cell physiology parameters in addition to CD89 expression. To pinpoint the CD89-dependency of the inflammatory responses, we isolated primary bone marrow derived cells (BMDC) from transgenic mice expressing the human CD89 on myeloid cells (CD89) and wildtype littermates (LM). Please note, that WT mice do not have a CD89 homolog (doi: 10.1073/pnas.94.10.5261; doi: 10.1074/jbc.272.11.7320). BMDC isolated from CD89+ mice and LM differ only in the presence or absence of CD89, providing a straightforward approach to identify CD89-dependent mechanisms. The original version already included IL-6 and TNF-alpha cytokine responses of BMDCs from CD89+ and LM mice upon exposure to uncoated and IgA-coated

bacterial extracellular vesicles (BEVs). Only CD89+ murine cells exposed to IgA-coated BEVs showed a pronounced pro-inflammatory cytokine response. This data is now provided in the new Fig. 6a and 6b.

In addition, we also provide data sets on BMDC isolated from CD89 and LM mice exposed to EV fractions from active UC patients and non-IBD controls (new Fig. 6c). When BMDCs isolated from LM were exposed to EV fractions from both groups (i.e. active UC patients and non-IBD controls) similar, low level cytokine responses were induced. Consistent with observations in CD89+CD14+CD11b+ U937 monocytes, EV fractions from active UC patients induced significantly higher IL-6 and TNF-alpha responses in CD89+ BMDCs compared to non-IBD controls. The use of BMDC isolated from CD89 mice also allowed the use of the commercially available Syk-inhibitor R406, which blocks the tyrosine kinase Syk representing a downstream factor in the CD89-dependent signaling pathway in CD89 mice (doi: 10.1038/s41385-019-0167-z.). Indeed, presence of the Syk-inhibitor significantly reduced the IL-6 and TNF-alpha cytokine responses evoked by IgA-coated BEVs as well as by EV samples from active UC patients (new Fig. 6b and d). The new data is provided in Fig. 6 and discussed in the appropriate sections of the manuscript as follows:

“As demonstrated recently, the tyrosine kinase Syk represents an essential downstream factor in the CD89-dependent signaling pathway in CD89 mice⁵¹. Treatment of CD89+ BMDCs with the Syk-inhibitor R406 significantly decreased IgA-coated BEVs^{B/L} mediated IL-6 and TNF-alpha responses (Fig. 6b). BMDC isolated from LM and CD89 mice are ideally suited to demonstrate CD89-dependent effects as they only differ in the expression of human CD89. Thus, we also assessed their cytokine responses upon exposure to seven randomly selected representative EV fractions from active UC patients and non-IBD controls (Fig. 6c). In BMDCs isolated from LM the EV fractions from both groups, i.e. active UC patients and non-IBD controls, induced similar, low level cytokine responses. Consistent with observations made with the CD89+CD14+CD11b+ U937 monocytes and primary human monocytes, EV fractions from active UC patients induced significantly higher IL-6 and TNF-alpha responses in CD89+ BMDCs compared to non-IBD controls. Treatment of CD89+ BMDCs with the Syk-inhibitor R406 significantly decreased IL-6 and TNF-alpha responses mediated by EV fractions from three randomly selected active UC patients (Fig. 6d). These results underpin the importance of CD89 including its downstream pathways for the strong pro-inflammatory responses evoked by IgA-coated BEVs^{B/L} as well as by the EV fractions from active UC patients.”

i) The authors state that mucosal inflammation is associated with IgA coating of bacterial MVs, which raises questions about the necessity of artificially introducing these vesicles during DSS-colitis experiments. If mucosal inflammation is indeed linked to the presence of IgA-coated bacterial MVs, one would expect these vesicles to already be present during DSS-colitis without the need for additional administration. Moreover, it is essential for the authors to consider whether these exogenously administered vesicles can effectively reach the gut and exert their intended effects. Given the complexities of gut physiology and the potential for vesicle clearance or degradation, it is crucial to label these vesicles to trace their biodistribution accurately.

Author response:

Please note that WT mice do not have a CD89 homolog (doi: 10.1073/pnas.94.10.5261; doi: 10.1074/jbc.272.11.7320) and mouse-IgA shows only weak binding to human CD89 (doi.org/10.1074/jbc.274.33.23508, doi.org/10.1084/jem.20112005; 10.1016/j.celrep.2019.03.062). Thus, even if DSS treatment results in increased coating of the murine gut microbiota and their vesicles with mouse-IgA, these coated vesicles would not result in strong CD89 activation in CD89-expressing mice nor in WT mice lacking a CD89-homolog. Therefore, the effect can only be observed in transgenic mice expressing the human CD89 receptor in combination with the administration of human IgA-coated BEVs, which is consistent with our results.

Since CD89+ mice show an exacerbation of inflammation upon oral gavage with human IgA-coated bacterial extracellular vesicles (BEVs) an effective amount must have reached the colon. Oral application of bacterial vesicles resulting in mucosal responses in the gut is well established in the field (summarized in doi: 10.7150/thno.85917; doi: 10.1128/mBio.01707-21; doi:10.1016/j.addr.2017.05.003; 10.3390/antibiotics12061045). Most importantly, a recent report detected high levels of vesicles derived from *Bacteroides* spp. in the mouse gut 8 h after their oral administration (10.3389/fmicb.2020.00057). Given such evidence in literature and the observed effects upon administration of IgA-coated BEVs in CD89-expressing mice, we think that undergoing massive experimental work to trace the biodistribution of BEVs is beyond the scope of this study. However, it is a line of investigation we would like to pursue in future research. Currently we thought it would be important to show the delivery of human IgA in mice receiving IgA-coated BEVs as this might undergo degradation by proteolysis. Thus, we analyzed cecal content collected from mice directly after sacrifice for human IgA by ELISA. Human IgA was only readily detected in mice receiving human IgA-coated BEVs, but not in mice receiving uncoated BEVs. Given

that the cecal content samples were collected approximately 24 h after the final gavage with vesicles, the presence of detectable human IgA clearly demonstrates that our effectors applied by oral gavage can reach the gut. We provide the new data sets in Suppl. Fig. S6d and describe the results in the appropriate section of the manuscript as follows:

“Human IgA was readily detectable in the cecal content of mice receiving IgA-coated BEVs^{B/L}, but not on mice receiving uncoated BEVs^{B/L} (Fig. S6d). Thus, the effectors applied by oral gavage have reached the relevant areas in the gut.”

Reviewer #3 (Remarks to the Author):

This study presents a compelling investigation showing that IgA-coated bacterial MVs elicit a proinflammatory response in a CD89-dependent manner. The findings contribute significantly to our understanding of the mechanisms underlying inflammatory responses in ulcerative colitis (UC) and offer potential insights for drug development. The study is conducted very well, employing a combination of in vitro and in vivo assays utilizing human patient samples and cell cultures, resulting in a convincing conclusion. I only have some minor comments:

1. In line 127 and elsewhere, it would be preferable to use the term "extracellular vesicles (EVs)" rather than "MV," for the human EV and bacterial MV mixtures, as "MV" typically refers to bacterial EVs, while EVs refer to a broader range of vesicles.

Author response:

We agree with the reviewer and changed the labelling as follows:

EV (extracellular vesicles) fractions for the pelleted human endoscopic fluid samples;

hEVs (host-derived extracellular vesicles) for the vesicles isolated from the human intestinal epithelial cell line HT-29;

BEVs (bacterial extracellular vesicles) for vesicles isolated from bacterial cultures [i.e. *Lactobacillus acidophilus*, *Bacteroides fragilis* and enterotoxigenic *Escherichia coli* (ETEC)]

2. In lines 143 to 144, please clarify what "%" refers to. It seems to indicate the percentage relative to the total protein biomass.

Author response:

Yes, it is percentage relative to the total protein biomass. We added the statement in the figure legend and explain the calculation in the respective method section.

3. In line 154, is there a specific reason for using only one bacterial species in the LTA assay compared to two bacteria in the LPS assay for the standard? While not critical, explaining any rationale behind this choice would be beneficial.

Author response:

For the quantification assays we used the bacterial extracellular vesicles (BEVs) derived from *Lactobacillus acidophilus*, *Bacteroides fragilis* and enterotoxigenic *Escherichia coli* (ETEC). As stated in the manuscript, "*L. acidophilus* and *B. fragilis* were chosen as (i) they represent highly abundant Gram-positive and -negative phyla present in the human gut microbiome^{48,49}, (ii) they can be cultured in liquid media with sufficient BEV-production, (iii) they are non-flagellated species excluding effects by flagellar components in the BEV preparations and (iv) a recent study reported that BEVs of these species elicited low inflammatory responses in intestinal cells⁵⁰. In contrast, ETEC BEVs originate from a flagellated, pathogenic bacterium and were among the most pro-inflammatory BEVs characterized⁵⁰."

This rationale led to BEVs from one Gram-positive and two Gram-negative representatives.

4. In Fig. 1a, is there a representative photo of non-IBD included? It is unclear what the blue box indicates for non-IBD.

Author response:

The revised version of Figure 1 now also includes a representative non-IBD image.

5. In Figs. 2 and 3, the MV fraction was dissolved in saline. Was a mock control (saline only) used for the MV fractions in the assays shown in Figs. 2 and 3?

Author response:

Yes, but cytokine levels in mock-treated control (saline only) were low or even below the limit of detection. In the revised version we provide mock-treated controls (saline only) for IL-8 and IL-6 detection in U937 (CD89⁺CD14⁺CD11b⁺) in Fig. 5a, 5b and S4b, for undifferentiated U937 cells in Fig. S4g-h as well as for IL-8 detection in HT-29 in Fig. S4i. The IL-6 and IL-8 responses shown in Fig. 2 and Fig. 3 (now Suppl. Fig. S5) are normalized to EV protein biomass (per μg) or to nanoparticle amount (per 10^{12} particles). Showing mock-treated control values in these graphs would require a division by 0 as the mock-treated control was exposed to 0 protein or 0 particles, which is mathematically not feasible.

6. In line 272, please provide further explanation for why ETEC MVs were used.

Author response:

We agree and extended our statement: "In contrast, ETEC BEVs originate from a flagellated, pathogenic bacterium and were among the most pro-inflammatory BEVs characterized ⁵⁰."

7. In Fig. 5, while I understand the rationale for using a mixture of *B. fragilis* and *L. acidophilus* MVs, are there any available data on their individual use? Including such information may provide valuable insights into whether the MVs have synergistic effects or not.

Author response:

We agree and exposed the U937 (CD89⁺CD14⁺CD11b⁺) cells to uncoated and IgA-coated BEVs derived from *Lactobacillus acidophilus* and *Bacteroides fragilis* individually. Please find the new data in Suppl. Fig. S4e. For both species, the IgA-coated BEVs induced significantly higher IL-8 and IL-6 responses in CD89⁺CD14⁺CD11b⁺ U937 monocytes compared to uncoated BEVs. We do not see a pronounced difference between BEVs from *Lactobacillus acidophilus* or *Bacteroides fragilis*. Thus, BEVs from both species seem to contribute to the cytokine responses.

We describe those results in the appropriate section: "Similar results were obtained by exposing the individual IgA-coated and uncoated BEVs from *L. acidophilus* or *B. fragilis* (BEVs^B and BEVs^L) to CD89⁺CD14⁺CD11b⁺ U937 monocytes (Fig. S4e) suggesting that IgA-coated BEVs from both species contribute to the cytokine response."

8. Typically, bacterial flagella are detected in MV fraction unless further purified. Therefore, stating that *B. fragilis* and *L. acidophilus* are non-flagellated bacteria, and at least MV fractions from these bacteria are devoid of flagella, could be important. In relation to this, were any flagella observed in the PEF fraction?

Author response:

We added a statement describing that these bacteria are non-flagellated. We did not specifically test for flagella in the PEF sample, but the TEM visualization of vesicles in the different preparations showed no obvious presence of flagellum-like structures (see new Suppl. Fig. S2). We have no direct evidence that such structures would be bacterial-derived and we cannot exclude that extracellular vesicles adhere differently to the TEM grids compared to sheared flagella pieces. Therefore, quantification on the basis of TEM is probably inaccurate. Based on these limitations we are not confident to report a quantitative amount of bacterial-derived flagella in these samples.

Moreover, the detected distribution including not closer definable matter is similar to a recently reported state-of-the-art protocol for MV isolation from human body fluids (doi: 10.1038/s41596-019-0236-5).

We added a statement on the potential contaminants:

"In addition to undetectable microbial vesicles, the remaining fraction could contain undigested food, intestinal mucus, fibers, flagella and pili. However, TEM analyses of human-derived EV fractions showed no obvious bacterial flagellar- and pili-like structures (Fig. S2). The distribution in the PEF samples is similar to a recently reported state-of-the-art protocol for EV isolation from human stool ²⁸."

9. Figs. 5c and d. It is shown that while uncoated MVs stimulated IL-6 and 8 release, they do not further stimulate cytokine production with increased dosage. In contrast, IgA-coated MVs induce a dose-dependent cytokine response. It may be worth discussing why such a difference in the dose-dependent manner occurs.

Author response:

We also consider this to be an interesting observation, but can only speculate on possible explanations. There are reports raising the possibility that interaction of IgA-immunocomplexes with CD89 stimulates phagocytosis. Therefore, the internalization of uncoated and IgA-coated BEVs in CD89⁺ cells is probably different. Maybe the amount of uncoated BEVs already reached saturation in the assays, while the IgA-coated BEVs utilize alternative routes, e.g. CD89-dependent pathways, and therefore are internalized at higher rates and/or amounts.

We added a statement on this thought.

We state: "The dose-independent cytokine response observed upon exposure to uncoated BEVs^{B/L} might indicate that saturation was reached with the BEV amounts tested. Notably, current reports highlight the possibility that interaction of IgA-immunocomplexes with CD89 stimulates phagocytosis³⁴. Thus, IgA-coated BEVs may utilize

alternative routes, e.g. CD89-dependent pathways, and therefore would be internalized at higher rates and amounts allowing the detection of dose-dependent cytokine responses.”

10. Regarding line 395, could there be further discussion on whether IgA-coated MVs or IgA-coated bacterial cells contribute to a larger extent to the proinflammatory response in UC?

Author response:

We agree and added a few thoughts.

We state: “Under homeostatic conditions IgA-opsonized bacteria are primarily restricted to the intestinal lumen, while under pathological conditions they can reach the lamina propria to interact with the immune cells^{59,60}. Similar principles may apply to BEVs, but whether IgA-coated bacteria or BEVs are more important in exacerbating inflammation remains to be elucidated. The direct visualization of BEVs and their differentiation from bacterial cells in the human intestine is limited by the available methods, as current light microscopy cannot visualize vesicle with a sufficiently high resolution. However, BEVs are non-living facsimiles of the donor cell vesicles and therefore share the same epitopes as the bacterial cell. Mechanistic studies on how BEVs spread *in vivo* are limited. However, it is believed that BEVs can pass the intestinal barrier more easily than bacteria and can deliver a concentrated dose of pro-inflammatory cargo, e.g. LPS, LTA, surface proteins or peptidoglycan^{15,61}.”

11. Line 547. It would be helpful to provide a reference or additional information that the vesicles are lysed with 0.1% SDS.

Author response:

Unfortunately, we found no report analyzing BEV lysis with SDS. According to general information on manufacturers websites 0.1 - 0.2% is sufficient to achieve proper cell lysis for bacteria such as *E. coli*. (e.g. <https://www.sigmaaldrich.com/deepweb/assets/sigmaaldrich/marketing/global/documents/134/298/cell-harvest-lysis-an7049en-mk.pdf>). Whether or not this holds true for BEVs remains to be elucidated. Notably, the study cited in our manuscript shows for a variety of BEVs derived from diverse Gram-positive and Gram-negative bacteria showed increased protein concentrations upon treatment with 0.1% SDS (see Table S4 of spectrum.01115-23-s0001.pdf). This indicates that the SDS treatment solubilizes at least some proteins, which would otherwise not be detected. Moreover, 0.1% SDS is the maximum concentration compatible with the Bradford assay. Thus, higher concentrations would not be feasible in the assay.

We added a statement in the method section to highlight this potential limitation:

“To allow detection of luminal content of vesicles samples were lysed with 0.1% SDS for 10 min prior to the assay. 0.1% SDS is the highest concentration compatible with the assay⁷¹. While addition of 0.1% SDS increases the amount of quantified proteins in a variety of EVs derived from diverse Gram-positive and Gram-negative bacteria⁵⁰, we cannot exclude insufficient lysis for some EV types present in the samples.”